# OpenFE: Automated Feature Generation beyond Expert-level Performance

## Abstract

The goal of automated feature generation is to liberate machine learning experts from the laborious task of manual feature generation, which is crucial for improving the learning performance of tabular data. The major challenge in automated feature generation is to efficiently and accurately identify useful features from a vast pool of candidate features. In this paper, we present OpenFE, an automated feature generation tool that provides competitive results against machine learning experts. OpenFE achieves efficiency and accuracy with two components: 1) a novel feature boosting method for accurately estimating the incremental performance of candidate features. 2) a feature-scoring framework for retrieving effective features from a large number of candidates through successive feature-wise halving and feature importance attribution. Extensive experiments on seven benchmark datasets show that OpenFE outperforms existing baseline methods. We further evaluate OpenFE in two famous Kaggle competitions with thousands of data science teams participating. In one of the competitions, features generated by OpenFE with a simple baseline model can beat 99.3% data science teams. In addition to the empirical results, we provide a theoretical perspective to show that feature generation is beneficial in a simple yet representative setting. Codes and datasets are in the supplementary materials.

## 1 Introduction

Feature generation is an important yet challenging task when applying machine learning methods to tabular data. Tabular data, where each row represents an instance and each column corresponds to a distinct feature, is ubiquitous in industrial applications and machine learning competitions. It has been well recognized that the quality of features has a significant impact on the learning performance of tabular data (Domingos, 2012). The goal of feature generation is to transform the base features into more informative ones to better describe the data and enhance the learning performance. For example, Price-to-Earnings ratio (P/E ratio), calculated as (share price)/(earnings per share), is derived from the base features "share price" and "earnings per share" in financial statements and informs investors about the value of a company. In practice, data scientists typically use their domain knowledge to find useful feature transformations in a trial-and-error manner, which requires tremendous human labor and expertise.

Since manual feature generation is time-consuming and requires case-by-case domain knowledge, automated feature generation emerges as an important topic in automated machine learning (Erickson et al., 2020; Lu, 2019). Expand-and-reduce is arguably the most prevalent framework in automated feature generation, in which we first expand the candidate features and then eliminate redundant ones (Kanter & Veeramachaneni, 2015; Lam et al., 2021; Kaul et al., 2017; Shi et al., 2020; Katz et al., 2016). There are two challenges in a typical expand-and-reduce practice. First, the number of candidate features is usually huge in many industrial applications. Calculating all candidate features is not only computationally expensive but also infeasible due to the enormous amount of memory required. The second challenge is how to efficiently and accurately estimate the incremental performance of a new feature, i.e., how much performance improvement a new candidate feature can offer when added to the base feature set. The majority of existing methods rely on statistical tests to determine if a new feature should be included (Kanter & Veeramachaneni, 2015; Lam et al., 2021; Shi et al., 2020). However, statistically significant features do not always translate into good predictors (Lo et al., 2015). Features may be significantly correlated with the target simply for a

small group of instances in the population, thereby leading to poor prediction in the population (Lo et al., 2015; Ward et al., 2010; Welch & Goyal, 2008). Besides, the effectiveness of a new feature may be encompassed by the base feature set, even if the new feature is significantly correlated with the target.

In this paper, we propose OpenFE, a powerful automated feature generation algorithm that can effectively generate useful features to enhance learning performance. First, motivated by the gap between significant features and good predictors, we propose a feature boosting method that directly estimates the predictive power of new features in addition to the base feature set. Second, inspired by the crucial fact that effective features are usually sparse in the huge number of candidate features, we propose a two-stage evaluation framework. In the first stage, we propose a successive featurewise pruning algorithm to quickly eliminate redundant candidate features by dynamically allocating computing resources to promising ones. In the second stage, we propose a feature importance attribution method to rank the remaining candidate features based on their contributions to the improvement in the learning performance and further eliminate redundant candidate features. We validate OpenFE on various datasets and Kaggle competitions, where OpenFE outperforms existing baseline methods. In a famous Kaggle competition with thousands of data science teams participating[1], the baseline model with features generated by OpenFE beats 99.3% of 6351 data science teams. More importantly, the features generated by OpenFE result in comparable or even larger performance improvement than those provided by the competition's top winners, demonstrating for the first time that automated feature generation is competitive against machine learning experts.

In addition to proposing a novel method, this paper intends to address two important problems that hinder the research process of automated feature generation. The first problem is that the majority of existing methods are evaluated on different datasets, and these studies do not open-source their codes and datasets, hindering new research from conducting fair comparisons. In order to facilitate fair comparisons in future research, we reproduce the main methods for automated feature generation and validate our reproduction by comparing the reproduced results with those in the corresponding papers. We will open-source the codes and datasets (see supplementary materials).

The second problem is the lack of evidence regarding the necessity of feature generation in the era of deep learning. Deep neural networks (DNNs) are widely recognized for their ability to extract feature representations. In recent years, a variety of DNNs have been carefully developed for modeling tabular data (Arık & Pfister, 2021; Gorishniy et al., 2021), and several of them have demonstrated their efficiency in feature interaction learning (Song et al., 2019; Wang et al., 2021). We extensively evaluate the effect of OpenFE on a variety of DNNs. We demonstrate that generating new transformed features with OpenFE can further enhance the learning performance of existing DNN architectures. In addition to the empirical results, we provide a theoretical justification of our feature generation procedure by presenting a simple yet representative transductive learning setting in which feature generation has provable benefits.

We summarize the contributions of our paper as follows:

- We propose a novel automated feature generation method that can effectively identify useful new features to enhance learning performance. Extensive experiments show that OpenFE achieves state-of-the-art on seven benchmark datasets. More importantly, we demonstrate for the first time that OpenFE is competitive against human experts in feature generation.
- We facilitate future research in feature generation by: 1) reproducing main methods for automated feature generation and releasing the codes and datasets, 2) providing empirical and theoretical evidence that feature generation is a crucial component in modeling tabular data, even in the era of deep learning.

## 2 PROBLEM DEFINITION

For a given training dataset $\mathcal{D}$, we split it into a sub-training set $\mathcal{D}_{tr}$ and a validation set $\mathcal{D}_{vld}$. Assume $\mathcal{D}$ consists of a feature set $\mathcal{T} + \mathcal{S}$, where $\mathcal{T}$ is the base feature set and $\mathcal{S}$ is the generated feature set. We use a learning algorithm $\mathcal{L}$ to learn a model $\mathcal{L}(\mathcal{D}_{tr}, \mathcal{T} + \mathcal{S})$, and compute the evaluation metric $\mathcal{E}(\mathcal{L}(\mathcal{D}_{tr}, \mathcal{T} + \mathcal{S}), \mathcal{D}_{vld}, \mathcal{T} + \mathcal{S})$ to measure the model performance, with a larger

---

[1]https://www.kaggle.com/competitions/ieee-fraud-detection/overview

value indicating better performance. Now, we formally define the feature generation problem as:

$$\max_{\mathcal{S} \subseteq A(\mathcal{T})} \mathcal{E}(\mathcal{L}(\mathcal{D}_{tr}, \mathcal{T} + \mathcal{S}), \mathcal{D}_{vld}, \mathcal{T} + \mathcal{S}) \tag{1}$$

where $A(\mathcal{T})$ is the set of all possible candidate features generated from the base feature set. The goal of feature generation is to find a feature set $\mathcal{S}$ from $A(\mathcal{T})$ that maximizes the evaluation metric. We determine $A(\mathcal{T})$ by the set $\mathcal{O}$ of operators we use to transform the base features. The operators $\mathcal{O}$ include unary operators, such as $log$, $sigmoid$, $square$, and binary operators, such as $\times$, $\div$, $min$, $max$, $GroupByThenMean$. Appendix D.2 contains the list of operators and how we use them to transform numerical and categorical features. Our method can be extended to include other operators, and users can also define their own operators.

## 3  OPENFE

### 3.1  OVERVIEW

We present an overview of OpenFE in Figure 1. OpenFE follows the expand-and-reduce framework for automated feature generation. For expansion, first, we classify all base features into numerical features and categorical features. Then we create a pool of candidate features by using operators to enumerate all the first-order transformations of base features, where each transformation uses one operator. The challenge of auto-mated feature generation often lies in re-

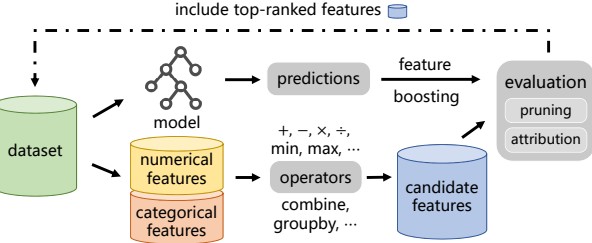

Figure 1: The overview of OpenFE.

duction after expansion, i.e., how to achieve both efficiency and effectiveness in eliminating redun-dant candidate features, especially when the number of candidate features is huge. For reduction, we propose a two-stage evaluation framework to quickly reduce the number of candidate features. Finally, we include top-ranked candidate features in the base feature set. We employ a greedy ap-proach and repeat the above procedure for high-order feature generation.

### 3.2  FEATURE BOOSTING

One of the key challenges in automated feature generation is to accurately estimate the incremental performance of a new feature, i.e., how much performance improvement the new feature can offer when added to the base feature set. A standard evaluation procedure involves including the new feature in the base feature set, retraining the machine learning model, and observing the change in validation loss (Katz et al., 2016). However, the standard evaluation procedure is prohibitively expensive due to the substantial computing resources required for training a model from scratch to convergence on the whole dataset. Therefore, many existing methods rely on statistical tests to determine if a new feature is effective (Christ et al., 2018; Kanter & Veeramachaneni, 2015; Shi et al., 2020). However, significant features do not always translate into good predictors (Lo et al., 2015). The effectiveness of a new feature may be encompassed by the base feature set. Motivated by this, we propose a feature boosting method to estimate the predictive power of new features in addition to the base feature set.

The inputs of feature boosting are a dataset $\mathcal{D}$, new features $\mathcal{T}'$, and base predictions $\hat{\boldsymbol{y}}$ (see the def-initions below). Feature boosting outputs the reduction in loss $\Delta$ as an estimate of the incremental performance of new features. Feature boosting is motivated by gradient boosting algorithms (Fried-man et al., 2000). Assume we have a dataset of $n$ samples $\mathcal{D} = \{(\boldsymbol{x}_i, y_i), i = 1, 2, ..., n\}$. Let $F$ denote a set of models defined over the base features $\mathcal{T}$. Assume we have a model $f \in F$ that has been trained to convergence on $\mathcal{D}$. We call the predictions $\hat{y}_i = f(\boldsymbol{x}_i)$ the *base predictions*, and define the loss $L(f) = \sum_{i=1}^{n} l(y_i, f(\boldsymbol{x}_i))$. Let $F'$ denote a set of models defined over new features $\mathcal{T}'$. We want to find a new model $f' \in F'$ to boost the performance of $f$ and minimize $L(f + f') = \sum_{i=1}^{n} l(y_i, f(\boldsymbol{x}_i) + f'(\boldsymbol{x}_i))$. We can fix $f$ and optimize $f'$ by gradient descent (see more details in Appendix D.3). The reduction in loss $\Delta = L(f) - L(f + f')$ is an estimate of the

incremental performance of new features. We call this method "feature boosting," which uses new features to boost the performance of base features. Initialized by base predictions, feature boosting can estimate incremental performance by training on new features as opposed to the full feature set, resulting in faster convergence for efficiency. We provide a case study in Appendix C.6 to illustrate that feature boosting can efficiently estimate the incremental performance of new features.

---

**Algorithm 1:** OpenFE

**input** : $\mathcal{D}$: dataset
**input** : $\mathcal{T}$: base features, $\mathcal{O}$: operators
**output:** New feature set.

Initialize order $\leftarrow 1$;
**while** *order < predefined max order* **do**
    Generate base predictions $\hat{\boldsymbol{y}}$ by $\mathcal{T}$ and $\mathcal{D}$
      using cross-validation;
    Enumerate candidate features $A(\mathcal{T})$ by $\mathcal{O}$
      and $\mathcal{T}$;
    # two-stage evaluation
    $A'(\mathcal{T}) = \text{SuccessivePruning}(A(\mathcal{T}), \mathcal{D}, \hat{\boldsymbol{y}})$;
    $A''(\mathcal{T}) = \text{FeatureAttribution}(A'(\mathcal{T}), \mathcal{D}, \hat{\boldsymbol{y}})$;
    $\mathcal{T} = \mathcal{T} + \text{top\_k}(A''(\mathcal{T}))$;
    order = order + 1;
return $\mathcal{T}$

**Algorithm 2:** Successive Pruning

**input** : $\mathcal{D}$: dataset, $\hat{\boldsymbol{y}}$: base predictions,
      $A(\mathcal{T})$: candidate feature set,
      $2^q$: #data blocks
**output:** Pruned new feature set.

Divide $\mathcal{D}$ equally into $2^q$ data blocks;
Initialize $A_0(\mathcal{T}) = A(\mathcal{T})$;
**for** $i \leftarrow 0$ **to** $q$ **do**
    **for** *new feature $\tau \in \mathcal{T}$* **do**
      Use a subset $\mathcal{D}_i$ with $2^i$ data blocks
        to calculate the values of $\tau$;
      Calculate the score of $\tau$ as
        $\Delta = \text{FeatureBoosting}(\mathcal{D}_i, \tau, \hat{\boldsymbol{y}})$;
    $A_i(\mathcal{T}) = \text{delete\_same}(A_i(\mathcal{T}))$;
    $A_{i+1}(\mathcal{T}) = \text{top\_half}(A_i(\mathcal{T}))$;
return $A_{q+1}(\mathcal{T})$

---

## 3.3 A Two-stage Evaluation Framework

Although feature boosting provides an efficient way to estimate the incremental performance of candidate features (defined in Section 3.1), it is still computationally infeasible to calculate and evaluate the huge number of candidate features. Inspired by a crucial fact that effective features are usually sparse in the vast pool of candidate features, we propose a two-stage evaluation framework based on feature boosting to efficiently eliminate redundant features and retrieve useful ones.

Before introducing the two-stage evaluation framework, we present the overall framework of OpenFE in Algorithm 1. OpenFE starts with generating the base predictions by training a model on the base features $\mathcal{T}$ for feature boosting. Then we create a pool of candidate features by applying operators on the base features. Next, we propose a two-stage evaluation framework to deal with the large number of candidate features. The first stage employs a successive featurewise pruning algorithm. By using a bandit-based algorithm (Jamieson & Talwalkar, 2016) to dynamically allocate computing resources to promising features, we quickly eliminate redundant features by examining the effectiveness of each feature alone using feature boosting. In the second stage, we further take interaction effects into account and propose a feature importance attribution algorithm to accredit the remaining features in reducing the loss function.

### 3.3.1 Stage I: Successive Featurewise Pruning

In OpenFE, first we use a successive featurewise pruning method to quickly reduce the number of candidate features (see Algorithm 2). The method is motivated by the successive halving algorithm in multi-armed bandit problems (Even-Dar et al., 2006; Zhou et al., 2014). Successive halving dynamically allocates computing resources to promising arms. In our settings, first we split the dataset into $2^q$ data blocks, where each data block has both a training set and a validation set, with a total of $\lfloor \frac{n}{2^q} \rfloor$ samples. Then we consider each candidate feature as an arm, and a pull of the arm is to use feature boosting to evaluate the effectiveness of the single feature on the data blocks. The reward of pulling an arm is the reduction in loss in feature boosting, which is calculated on the validation set. We allocate more data blocks (resources) to more promising candidate features. After successive featurewise pruning, only candidate features with a positive reduction in loss $\Delta$ are returned.

Besides successive halving, we use a simple trick in our algorithm to eliminate redundant candidate features. It is common in the candidate feature set for two features to have exactly the same values.

For example, $max(\tau_1, \tau_2)$ and $max(\tau_1, \tau_3)$ are exactly the same if the minimum value of feature $\tau_1$ is larger than the maximum value of $\tau_2$ and $\tau_3$. Features with the same values have the same score (i.e., reduction in loss $\Delta$). Therefore, the $delete\_same$ algorithm first ranks all the new features by their scores, and then compares the scores of adjacent features to remove redundant ones.

### 3.3.2 STAGE II: FEATURE IMPORTANCE ATTRIBUTION

There are two problems unaddressed in stage I: 1) There are still many redundant features in the candidate feature set after pruning. 2) Stage I only evaluates the effectiveness of each feature alone. Therefore, in stage II, we further consider the interaction effects to evaluate the effectiveness of the remaining candidate features. We select the top-ranked candidate features and remove redundant ones to improve generalization performance. Assume the remaining candidate features after pruning is $A'(\mathcal{T})$. We use candidate features $A'(\mathcal{T})$ and base features $\mathcal{T}$ together as the inputs to feature boosting. The

---

**Algorithm 3:** Feature Attribution

**input :** $\mathcal{D}$: dataset, $\hat{y}$: base predictions,
$\quad\quad\quad$ $\mathcal{T}$: base features,
$\quad\quad\quad$ $A'(\mathcal{T})$: candidate feature set
**output:** Sorted $A'(\mathcal{T})$.

$\Delta = \text{FeatureBoosting}(\mathcal{D}, \mathcal{T} + A'(\mathcal{T}), \hat{y})$;
Attribute the reduction in loss $\Delta$ to each
$\quad$ feature in $A'(\mathcal{T})$ as their importance;
Sort $A'(\mathcal{T})$ according to importance;
return sorted $A'(\mathcal{T})$

---

reduction in loss $\Delta$ is an estimate of the incremental performance of the remaining candidate features $A'(\mathcal{T})$, which considers the interaction effects between $\mathcal{T}$ and $A'(\mathcal{T})$. We attribute the reduction in loss $\Delta$ to each candidate feature as their importance. Popular methods for feature importance attribution include mean decrease in impurity (MDI) (Breiman, 2001), permutation feature importance (PFI) (Breiman, 2001), and SHAP (Lundberg et al., 2018). MDI is a popular method for feature attribution in tree ensembles, which sums up the total reduction of loss in all splits for a given feature. PFI measures the increase in the loss function after we randomly permute the feature's values. SHAP is a game-theoretic method for feature attribution.

### 3.4 IMPLEMENTATION

In OpenFE, we use gradient boosting decision trees (GBDT) (Friedman, 2001) to model tabular data for two reasons: 1) GBDT is usually the best performing model on tabular data where features are individually meaningful (Gorishniy et al., 2021; Borisov et al., 2021). 2) GBDT can automatically handle missing values and categorical features, which is convenient for automation (Ke et al., 2017). We use the popular LightGBM implementation (Ke et al., 2017). We use MDI for feature importance attribution. We compare MDI, PFI, and SHAP in Appendix C.4. We show that MDI is fast to compute and provides comparable performance to PFI and SHAP. Even though the feature generation method relies on GBDT, the generated features can also enhance the learning performance of a variety of DNNs (see Section 5.4).

## 4 THEORETICAL ADVANTAGE OF FEATURE GENERATION

In this section, we study the advantage of feature generation from a theoretical perspective. We present a simple yet representative setting in which the test loss of empirical risk minimization augmented with feature generation converges to zero provably as the number of training samples increases, while the test loss for any learning model without feature generation is at least a positive constant. In particular, we present a transductive learning setting, which captures important characteristics of a class of datasets one may encounter frequently in data science applications, e.g., the IEEE-CIS Fraud Detection dataset (we also conduct experiments on this dataset in Section 5.5). Due to space limit, a formal and detailed description of the model can be found in Appendix A. We briefly introduce the high-level idea here.

Many tabular datasets contain both categorical and numerical attributes (i.e., features). A categorical feature partitions the dataset into groups (each associated with a distinct category). For a data point $(X, Y)$. the target $Y$ is correlated with not only the feature $X$, but also certain statistics of the group containing $(X, Y)$. Datasets with such characteristics are abundant in data science applications. As a concrete example, one may think each training data as a transaction, and one categorical feature is user id (each user may have many transactions in this table. These transactions form a group). The

target $Y$ we want to predict about the transaction (e.g., probability of fraudulence) may depend on not only the features of this particular transaction, but also some statistics of this user (e.g., average size of his/her transactions). Hence, one can see that operations such as $\mathrm{GroupByThenMean}$ (group by the user id) can provide statistical information about the user by aggregating the information from all data points associated to this user.

We present a theoretical data model, which is a two-phase data generation model, to capture the above characteristics. Under fairly standard learning theoretic assumptions (i.e., bounded Rademacher complexity), we prove that empirical risk minimization augmented with feature generation (such as the $\mathrm{GroupByThenMean}$ operation) can achieve vanishing test loss as the sample size and group size increase.

**Theorem 1.** *(informal) Assume the data set is generated according to the two-phase process described in Appendix A. Denote the number of groups in the training set and test set by $k_1$ and $k_2$ respectively, and the number of data points in each group by $h$. There is a feature generation function $H$, such that the test loss of the empirical risk minimizer $\hat{f}$ can be bounded by*

$$L_{\mathcal{D}_{test}}(H, \hat{f}) \leq O\left(\mathrm{Rad}_{k_1}(\mathcal{F}) + \sqrt{\ln(4\delta^{-1})/k_1} + \sqrt{d\ln(4d(k_1+k_2)\delta^{-1})/h}\right)$$

*with probability at least $1 - \delta$. In particular, assuming the Rademacher complexity $\mathrm{Rad}_{k_1}(\mathcal{F}) \to 0$ as $k_1 \to \infty$, the test loss approaches to $0$ when $k_1, h \to \infty$.*

On the other hand, if we do not use any feature generation, we prove that any predictor $f'$ (no matter how complicated $f'$ is) incurs a non-vanishing constant test loss.

**Theorem 2.** *(informal) In case that we do not use any feature generation, there exists a problem instance such that, no matter how large $k_1, k_2$, and $h$ are, for any predictor $f' : \mathcal{X} \to \mathcal{Y}$, the test loss $L_{\mathcal{D}_{test}}(f') \geq \frac{3}{64}$.*

## 5 EXPERIMENTS

### 5.1 DATASETS AND EVALUATION METRICS

We use a diverse set of seven public datasets with two regression datasets, three binary classification datasets, and two multi-class classification datasets. Each dataset has exactly one train-validation-test split, and all methods use the same split. The datasets we collect include: California Housing (CA, real estate data, (Pace & Barry, 1997)),

Table 1: Properties of datasets used in our experiments. Notation: "RMSE" $\sim$ root-mean-square error, "AUC" $\sim$ area-under-curve, "Acc." $\sim$ accuracy.

|  | CA | MI | DI | NO | VE | JA | CO |
|---|---|---|---|---|---|---|---|
| #objects | 20640 | 1200192 | 101766 | 34465 | 98528 | 83733 | 581012 |
| #num. features | 7 | 111 | 3 | 34 | 100 | 54 | 9 |
| #cat. features | 0 | 0 | 34 | 29 | 0 | 0 | 0 |
| #ord. features | 1 | 25 | 10 | 55 | 0 | 0 | 45 |
| metric | RMSE | RMSE. | AUC | AUC | AUC | Acc. | Acc. |
| #classes | – | – | 2 | 2 | 2 | 4 | 7 |

Microsoft (MI, search queries, (Qin & Liu, 2013)), Diabetes (DI, hospital records, (Strack et al., 2014)), Nomao (NO, data deduplication, (Candillier & Lemaire, 2012)), Vehicle (VE, vehicle classification, (Siebert, 1987)), Jannis (JA, anonymized dataset, (Guyon et al., 2019)), Covertype (CO, forest characteristics, (Blackard & Dean, 1999)). We summarize the dataset properties in Table 1. The features in each dataset are classified into numerical, categorical, and ordinal features. An ordinal feature can be treated as both categorical or numerical when generating transformed features.

### 5.2 BASELINE METHODS FOR COMPARISONS

The baseline methods for comparisons include: **Base** (the base feature set without feature generation), **FCTree** (Fan et al., 2010), **SAFE** (Shi et al., 2020), **AutoFeat** (Horn et al., 2019), **AutoCross** (Luo et al., 2019). We provide another baseline that uses the last hidden layer of DCN-V2 as new features. DCV-V2 (Wang et al., 2021) is a DNN architecture developed for modeling tabular data and claims to be able to capture feature interactions automatically. We denote this baseline as **NN**. Most existing automated feature engineering methods do not open-source their codes. We reproduce FCTree, SAFE, and AutoCross according to the descriptions in these paper. We validate our reproduction by comparing the reproduced results with those in the corresponding papers

(see Appendix C.2). Some other learning-based methods, such as LFE (Nargesian et al., 2017), ExploreKit (Katz et al., 2016) (meta learning) and NFS (Chen et al., 2019), TransGraph (Khurana et al., 2018) (reinforcement learning), lack critical details for reproduction. For example, the choice of training datasets is crucial for generalization in learning-based methods, however previous research did not describe which datasets were utilized for training (Nargesian et al., 2017; Khurana et al., 2018). We run OpenFE on the test datasets they use and compare our results with the numbers reported in their papers in Appendix C.1.

Table 2: Comparisons between OpenFE and baseline methods. We report the mean and standard deviation of the evaluation metric on the test set over 10 random seeds. The top result for each dataset is marked in **bold**. SAFE and AutoCross can only run on binary-classification datasets. See Table 1 for the evaluation metric of different tasks.

| Method | CA ↓ | MI ↓ | DI ↑ | NO ↑ | VE ↑ | JA ↑ | CO ↑ |
|---|---|---|---|---|---|---|---|
| | | | | LightGBM | | | |
| Base | $0.4323_{\pm 3.7e\text{-}3}$ | $0.7439_{\pm 3.0e\text{-}4}$ | $0.7313_{\pm 1.2e\text{-}3}$ | $0.9958_{\pm 2.0e\text{-}4}$ | $0.9250_{\pm 3.0e\text{-}4}$ | $0.7212_{\pm 1.9e\text{-}3}$ | $0.9685_{\pm 1.1e\text{-}3}$ |
| NN | $0.4791_{\pm 1.3e\text{-}3}$ | $0.7496_{\pm 1.0e\text{-}4}$ | $0.7174_{\pm 1.0e\text{-}3}$ | $0.9924_{\pm 5.0e\text{-}4}$ | $0.9241_{\pm 6.0e\text{-}4}$ | $0.7195_{\pm 8.0e\text{-}4}$ | $0.9659_{\pm 2.0e\text{-}4}$ |
| FCTree | $0.4324_{\pm 2.7e\text{-}3}$ | $0.7440_{\pm 1.0e\text{-}4}$ | $0.7313_{\pm 1.0e\text{-}3}$ | $0.9956_{\pm 2.0e\text{-}4}$ | $0.9255_{\pm 2.0e\text{-}4}$ | $0.7190_{\pm 1.3e\text{-}3}$ | $0.9710_{\pm 7.0e\text{-}4}$ |
| SAFE | – | – | $0.7295_{\pm 1.7e\text{-}3}$ | $0.9958_{\pm 3.0e\text{-}4}$ | $0.9250_{\pm 1.0e\text{-}3}$ | – | – |
| AutoFeat | $0.4440_{\pm 2.5e\text{-}3}$ | $0.7438_{\pm 2.0e\text{-}4}$ | $0.7318_{\pm 1.1e\text{-}3}$ | $0.9958_{\pm 2.0e\text{-}4}$ | $0.9250_{\pm 3.0e\text{-}4}$ | $0.7205_{\pm 7.0e\text{-}4}$ | $0.9682_{\pm 1.1e\text{-}3}$ |
| AutoCross | – | – | $0.7319_{\pm 1.1e\text{-}3}$ | $0.9932_{\pm 3.0e\text{-}4}$ | $0.9207_{\pm 3.0e\text{-}4}$ | – | – |
| OpenFE | $\mathbf{0.4206_{\pm 1.6e\text{-}3}}$ | $\mathbf{0.7382_{\pm 3.0e\text{-}4}}$ | $\mathbf{0.8878_{\pm 2.5e\text{-}3}}$ | $\mathbf{0.9967_{\pm 2.0e\text{-}4}}$ | $\mathbf{0.9281_{\pm 4.0e\text{-}4}}$ | $\mathbf{0.7286_{\pm 1.4e\text{-}3}}$ | $\mathbf{0.9737_{\pm 5.0e\text{-}4}}$ |
| | | | | FT-Transformer | | | |
| Base | $0.4590_{\pm 3.3e\text{-}3}$ | $0.7461_{\pm 4.0e\text{-}4}$ | $0.7281_{\pm 2.4e\text{-}3}$ | $0.9933_{\pm 5.0e\text{-}4}$ | $0.9272_{\pm 6.0e\text{-}4}$ | $0.7328_{\pm 2.3e\text{-}3}$ | $0.9696_{\pm 8.0e\text{-}4}$ |
| NN | $0.4797_{\pm 9.0e\text{-}4}$ | $0.7503_{\pm 2.0e\text{-}4}$ | $0.7112_{\pm 4.0e\text{-}4}$ | $0.9921_{\pm 4.0e\text{-}4}$ | $0.9246_{\pm 2.0e\text{-}4}$ | $0.7165_{\pm 1.5e\text{-}3}$ | $0.9663_{\pm 9.0e\text{-}4}$ |
| FCTree | $0.4570_{\pm 2.7e\text{-}3}$ | $0.7461_{\pm 4.0e\text{-}4}$ | $0.7282_{\pm 1.4e\text{-}3}$ | $0.9930_{\pm 5.0e\text{-}4}$ | $0.9273_{\pm 4.0e\text{-}4}$ | $0.7298_{\pm 2.4e\text{-}3}$ | $\mathbf{0.9713_{\pm 7.0e\text{-}4}}$ |
| SAFE | – | – | $0.7281_{\pm 1.1e\text{-}3}$ | $0.9933_{\pm 4.0e\text{-}4}$ | $0.9276_{\pm 3.0e\text{-}3}$ | – | – |
| AutoFeat | $0.4574_{\pm 3.0e\text{-}3}$ | $0.7463_{\pm 3.0e\text{-}4}$ | $0.7281_{\pm 2.6e\text{-}3}$ | $0.9931_{\pm 4.0e\text{-}4}$ | $0.9271_{\pm 5.0e\text{-}4}$ | $0.7337_{\pm 1.2e\text{-}3}$ | $0.9704_{\pm 7.0e\text{-}4}$ |
| AutoCross | – | – | $0.7292_{\pm 9.0e\text{-}4}$ | $0.9905_{\pm 3.0e\text{-}4}$ | $0.9205_{\pm 3.0e\text{-}4}$ | – | – |
| OpenFE | $\mathbf{0.4530_{\pm 2.7e\text{-}3}}$ | $\mathbf{0.7411_{\pm 7.0e\text{-}4}}$ | $\mathbf{0.8870_{\pm 1.2e\text{-}3}}$ | $\mathbf{0.9950_{\pm 3.0e\text{-}4}}$ | $\mathbf{0.9289_{\pm 3.0e\text{-}4}}$ | $\mathbf{0.7379_{\pm 2.3e\text{-}3}}$ | $0.9708_{\pm 7.0e\text{-}4}$ |

## 5.3 COMPARISON RESULTS

In this section, we compare OpenFE with other baseline methods. We use two standard learning algorithms to evaluate the effectiveness of the new features generated by different methods. For GBDT, we choose the LightGBM implementation (Ke et al., 2017). For neural networks, we choose FT-Transformer (Gorishniy et al., 2021). Hyperparameter tuning follows a standard benchmark study (Gorishniy et al., 2021), and we tune the hyperparameters using the base feature set. For each dataset, we generate first-order features and include the same number of new features for different methods for a fair comparison. We discuss generating high-order features in Appendix E.2. We present the comparison results in Table 2. We can observe that OpenFE has clear advantages over baseline methods and achieves SOTA in most cases. The features generated by OpenFE greatly enhance the performance of both LightGBM and FT-Transformer. When baseline methods fail to generate effective new features, they include uninformative features in the base feature set, which brings additional noise and does harm to the generalization of the learning model. We present the running time of different methods in Appendix C.3.

## 5.4 FEATURE GENERATION FOR NEURAL NETWORKS

In this section, we show that new features generated by OpenFE can greatly improve the performance of a variety of neural networks designed specifically for tabular data. The models include: **AutoInt** (Song et al., 2019), **DCN-V2** (Wang et al., 2021), **NODE** (Popov et al., 2019), **TabNet** (Arık & Pfister, 2021), **FT-Transformer** (Gorishniy et al., 2021). We follow the same implementations and hyperparameter tuning of these networks in (Gorishniy et al., 2021). We present the results in Table 3. The features generated by OpenFE greatly enhance the performance of different models in most cases. For AutoInt and DCN-V2 which claim to be able to learn feature interactions, feature generation can further improve the performance. Even though OpenFE relies on GBDT to measure the performance of new features, the generated features are also effective for a variety of DNNs.

Table 3: The effect of OpenFE on a variety of DNNs. We report the mean and standard deviations of the evaluation metric on the test set over 10 random seeds. The top results are shown in **bold**.

| Model | Features | CA $\downarrow$ | MI $\downarrow$ | NO $\uparrow$ | VE $\uparrow$ | JA $\uparrow$ | CO $\uparrow$ |
|---|---|---|---|---|---|---|---|
| AutoInt | Base | $0.4741_{\pm 3.7e\text{-}3}$ | $0.7501_{\pm 5.0e\text{-}4}$ | $0.9927_{\pm 3.0e\text{-}4}$ | $0.9270_{\pm 9.0e\text{-}4}$ | $0.7207_{\pm 2.3e\text{-}3}$ | $0.9342_{\pm 4.2e\text{-}3}$ |
| | OpenFE | $0.4618_{\pm 5.1e\text{-}3}$ | $0.7447_{\pm 4.0e\text{-}4}$ | $0.9943_{\pm 4.0e\text{-}4}$ | $0.9287_{\pm 7.0e\text{-}4}$ | $0.7243_{\pm 2.7e\text{-}3}$ | $0.9415_{\pm 4.1e\text{-}3}$ |
| DCN-V2 | Base | $0.4832_{\pm 2.1e\text{-}3}$ | $0.7486_{\pm 7.0e\text{-}4}$ | $0.9926_{\pm 4.0e\text{-}4}$ | $0.9236_{\pm 1.5e\text{-}3}$ | $0.7158_{\pm 1.6e\text{-}3}$ | $0.9653_{\pm 1.3e\text{-}3}$ |
| | OpenFE | $0.4722_{\pm 4.0e\text{-}3}$ | $0.7431_{\pm 4.0e\text{-}4}$ | $0.9944_{\pm 1.0e\text{-}4}$ | $0.9260_{\pm 4.0e\text{-}4}$ | $0.7163_{\pm 1.6e\text{-}3}$ | $0.9633_{\pm 7.0e\text{-}4}$ |
| TabNet | Base | $0.5096_{\pm 4.9e\text{-}3}$ | $0.7508_{\pm 1.1e\text{-}3}$ | $0.9932_{\pm 3.0e\text{-}4}$ | $0.9236_{\pm 1.0e\text{-}3}$ | $0.7237_{\pm 4.3e\text{-}3}$ | $0.9594_{\pm 4.3e\text{-}3}$ |
| | OpenFE | $0.5011_{\pm 9.0e\text{-}3}$ | $0.7464_{\pm 8.0e\text{-}4}$ | $0.9948_{\pm 2.0e\text{-}4}$ | $0.9258_{\pm 7.0e\text{-}4}$ | $0.7237_{\pm 4.3e\text{-}3}$ | $0.9592_{\pm 1.4e\text{-}3}$ |
| NODE | Base | $0.4639_{\pm 1.6e\text{-}3}$ | $0.7454_{\pm 3.0e\text{-}4}$ | $0.9937_{\pm 4.0e\text{-}4}$ | $0.9265_{\pm 3.0e\text{-}4}$ | $0.7271_{\pm 1.8e\text{-}3}$ | $0.9589_{\pm 1.2e\text{-}3}$ |
| | OpenFE | $0.4569_{\pm 1.8e\text{-}3}$ | $\mathbf{0.7395_{\pm 2.0e\text{-}4}}$ | $\mathbf{0.9953_{\pm 5.0e\text{-}4}}$ | $\mathbf{0.9289_{\pm 4.0e\text{-}4}}$ | $0.7312_{\pm 1.1e\text{-}3}$ | $0.9632_{\pm 9.0e\text{-}4}$ |
| FT-Transformer | Base | $0.4590_{\pm 3.3e\text{-}3}$ | $0.7461_{\pm 4.0e\text{-}4}$ | $0.9933_{\pm 5.0e\text{-}4}$ | $0.9272_{\pm 6.0e\text{-}4}$ | $0.7328_{\pm 2.3e\text{-}3}$ | $0.9696_{\pm 8.0e\text{-}4}$ |
| | OpenFE | $\mathbf{0.4530_{\pm 2.7e\text{-}3}}$ | $0.7411_{\pm 7.0e\text{-}4}$ | $0.9950_{\pm 3.0e\text{-}4}$ | $\mathbf{0.9289_{\pm 3.0e\text{-}4}}$ | $\mathbf{0.7379_{\pm 2.3e\text{-}3}}$ | $\mathbf{0.9708_{\pm 7.0e\text{-}4}}$ |

Table 4: Results of OpenFE and Expert (feature generation by experts) in two Kaggle competitions. Notation: pub. $\sim$ public, pri. $\sim$ private. The final standing is determined according to the scores in the private leaderboard.

| feature | order | IEEE-CIS Fraud Detection | | | | BNP Paribas Cardif Claims Management | | | |
|---|---|---|---|---|---|---|---|---|---|
| | | pub. score | pub. rank | pri. score | pri. rank | pub. score | pub. rank | pri. score | pri. rank |
| Base | - | $0.9463_{\pm 3.7e\text{-}4}$ | 2408/6351 | $0.9182_{\pm 3.2e\text{-}4}$ | 2286/6351 | $0.4382_{\pm 1.7e\text{-}4}$ | 38/2920 | $0.4359_{\pm 2.0e\text{-}4}$ | 31/2920 |
| Expert | first | $0.9602_{\pm 2.6e\text{-}4}$ | 52/6351 | $0.9327_{\pm 3.0e\text{-}4}$ | 70/6351 | $0.4334_{\pm 1.7e\text{-}4}$ | 14/2920 | $0.4308_{\pm 1.0e\text{-}4}$ | 12/2920 |
| | high | $0.9597_{\pm 3.1e\text{-}4}$ | 54/6351 | $0.9320_{\pm 3.3e\text{-}4}$ | 76/6351 | $0.4322_{\pm 1.5e\text{-}4}$ | 13/2920 | $0.4301_{\pm 2.2e\text{-}4}$ | 12/2920 |
| OpenFE | first | $0.9617_{\pm 3.5e\text{-}4}$ | 38/6351 | $0.9360_{\pm 4.0e\text{-}4}$ | 44/6351 | $0.4349_{\pm 1.4e\text{-}4}$ | 22/2920 | $0.4324_{\pm 2.9e\text{-}4}$ | 16/2920 |
| | high | $0.9617_{\pm 3.6e\text{-}4}$ | 38/6351 | $0.9363_{\pm 2.8e\text{-}4}$ | 42/6351 | $0.4324_{\pm 2.0e\text{-}4}$ | 14/2920 | $0.4302_{\pm 3.2e\text{-}4}$ | 12/2920 |

## 5.5 CASE STUDIES: COMPARING OPENFE WITH MACHINE LEARNING EXPERTS

We present two Kaggle competitions to show that the features generated by OpenFE can result in competitive performance improvement against those provided by the competition's top winners.

The first Kaggle competition is IEEE-CIS Fraud Detection[2], where the goal is to predict whether an online transaction is fraudulent. This competition is one of the largest and most competitive tabular data competitions on Kaggle, with 6,351 data science teams participating. The competition's first place team made public the features they generated after the competition ended[3], which we refer to as Expert. This competition heavily relies on feature generation, where a baseline model of XGBoost without feature generation ranks at 2286 among 6351

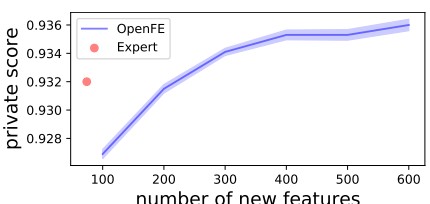

Figure 2: Comparison between Expert and OpenFE.

teams on the private leaderboard. However, the baseline model with features generated by the team ranks at 76/6351. The same baseline model with features generated by OpenFE ranks at 42/6351, which outperforms the features generated by the first-place team. We present the results in Table 4. We investigate the influence of the number of new first-order features in Figure 2. The advantage of OpenFE is that it can automatically explore and generate more useful feature transformations than Expert. We explain the useful feature transformations generated by OpenFE in Appendix E.1.

The second competition is BNP Paribas Cardif Claims Management[4], where the goal is to predict whether an insurance claim should be approved. We present the details in Appendix C.5.

---

[2] https://www.kaggle.com/competitions/ieee-fraud-detection/overview

[3] https://www.kaggle.com/code/cdeotte/xgb-fraud-with-magic-0-9600

[4] https://www.kaggle.com/competitions/bnp-paribas-cardif-claims-management/

Table 5: Results for ablation study. The model is LightGBM.

| | CA ↓ | MI ↓ | DI ↑ | NO ↑ | VE ↑ | JA ↑ | CO ↑ |
|---|---|---|---|---|---|---|---|
| OpenFE | $0.4206_{\pm 1.6e\text{-}3}$ | $0.7382_{\pm 3.0e\text{-}4}$ | $0.8878_{\pm 2.5e\text{-}3}$ | $0.9967_{\pm 2.0e\text{-}4}$ | $0.9281_{\pm 4.0e\text{-}4}$ | $0.7286_{\pm 1.4e\text{-}3}$ | $0.9737_{\pm 5.0e\text{-}4}$ |
| OpenFE\FB | $0.4299_{\pm 2.2e\text{-}3}$ | $0.7441_{\pm 4.0e\text{-}3}$ | $0.8856_{\pm 3.3e\text{-}3}$ | $0.9959_{\pm 2.0e\text{-}4}$ | $0.9263_{\pm 2.0e\text{-}4}$ | $0.7251_{\pm 1.2e\text{-}3}$ | $0.9704_{\pm 7.0e\text{-}4}$ |
| OpenFE\SS | $0.4282_{\pm 2.3e\text{-}3}$ | $0.7439_{\pm 5.0e\text{-}4}$ | $0.8846_{\pm 2.2e\text{-}3}$ | $0.9952_{\pm 3.0e\text{-}4}$ | $0.9267_{\pm 3.0e\text{-}4}$ | $0.7316_{\pm 1.1e\text{-}3}$ | $0.9649_{\pm 7.0e\text{-}4}$ |
| OpenFE-MI | $0.4282_{\pm 2.3e\text{-}3}$ | $0.7439_{\pm 3.0e\text{-}4}$ | $0.8867_{\pm 4.4e\text{-}3}$ | $0.9959_{\pm 1.0e\text{-}4}$ | $0.9261_{\pm 3.0e\text{-}4}$ | $0.7293_{\pm 1.2e\text{-}3}$ | $0.9713_{\pm 5.0e\text{-}4}$ |

## 5.6 ABLATION STUDY

We conduct an ablation study to justify the design choices of OpenFE. We name different variants of OpenFE as follows:

- OpenFE\FB. OpenFE without feature boosting. We train GBDT from scratch without using base predictions as the initial prediction. For example, in regression tasks the initial prediction is the average of the targets in the training set.
- OpenFE\SS. OpenFE without successive featurewise pruning. We subsample the data so that generating all the features can fit in the memory.
- OpenFE-MI. We use mutual information between the feature and the target as the scoring criterion in successive featurewise pruning.

We present the results in Table 5. The results show that: 1) Feature boosting significantly improves the results. 2) Directly subsampling the data usually hurts the performance. 3) Mutual information does not provide desirable results in eliminating features.

## 6 RELATED WORK

Expand-and-reduce is arguably the most popular framework in automated feature generation. Most existing automated feature generation approaches employ statistical methods to identify and remove redundant features. For example, Data Science Machine (DSM) (Kanter & Veeramachaneni, 2015) uses the f-value between the feature and the target, and One Button Machine (OneBM) (Lam et al., 2021) uses the Chi-square test to remove redundant features. DSM and OneBM focus on feature generation in relational databases, which is different from our setting in tabular data. FC-Tree (Fan et al., 2010) uses information gain to select new features during the construction of decision trees. SAFE (Shi et al., 2020) uses information value to exclude uninformative features. AutoCross (Luo et al., 2019) and AutoFeat (Horn et al., 2019) uses the improvement on a linear regression model to determine whether a new feature should be included. LFE (Nargesian et al., 2017) and ExploreKit (Katz et al., 2016) design meta features based on statistical methods and use a meta learning approach to determine the effectiveness of new features. LFE and ExploreKit are confined to binary-classification tasks and a specific set of operators. How to search for high-order feature transformations is challenging in automated feature generation due to the explosion in search space (Chen et al., 2019). Some learning-based methods directly search for high-order feature transformations (Xie et al., 2021). For example, Neural Feature Search (Chen et al., 2019) utilizes a recurrent neural network controller to search for a series of transformations. Khurana et al. (2018) uses reinforcement learning to traverse a transformation graph for high-order features.

## 7 CONCLUSION

We propose OpenFE, a powerful automated feature generation that can effectively generate useful features to enhance the learning performance of tabular data. Extensive experiments show that OpenFE achieves SOTA on seven benchmark datasets and is competitive against human experts in feature generation. We facilitate future research by: 1) reproducing the main methods for automated feature generation and releasing the codes and datasets for fair comparisons. 2) providing empirical and theoretical evidence that feature generation is a crucial component in modeling tabular data.

## REPRODUCIBILITY STATEMENT

All the codes and datasets can be found in the supplementary materials, including the implementation of OpenFE and other baseline methods. We also provide scripts (detailed experiment parameters) in the supplementary materials to reproduce the experiments. Readers can follow the "readme" to download the datasets and reproduce the results. Our theoretical contributions are clearly stated in Appendix A with definitions, assumptions, and complete proofs of all the theorems.

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

# A    THEORETICAL RESULTS

PROBLEM SETTING

In this section, we study the advantage of feature generation from a theoretical perspective. In particular, we present a simple yet representative setting in which empirical risk minimization augmented with feature generation can achieve smaller test loss than any learning model without feature generation.

Consider a regression problem with both numerical and categorical features in a transductive learning setting. Denote $\mathcal{X}^{\text{num}} \subseteq \mathbb{R}^d$ as the numerical feature space, $\mathcal{X}^{\text{cat}} \subseteq \mathbb{N}$ the categorical feature space, and $\mathcal{Y} \subseteq [0,1]$ as the target domain. We assume $\mathcal{X}^{\text{num}}$ is $d$-dimensional and convex. The training data $\mathcal{D}_{\text{train}}$ consists of $n$ training data $\{(X_i, Y_i)\}_{i=1,\ldots,n}$ where $X_i = (x_{i0}, x_{i1}, \ldots, x_{id})$, $x_{i0} \in \mathcal{X}^{\text{cat}}$ is the categorical feature value, $(x_{i1}, \ldots, x_{id}) \in \mathcal{X}^{\text{num}}$ are $d$ numerical feature values. As a concrete example, one may think each training data as a transaction, where $x_{i0}$ as the user id and $(x_{i1}, \ldots, x_{id})$ are some features about the user and the transaction, and $Y$ is the target we want to predict about the transaction (e.g., lateness of payment, probability of fraudulence). The test data $\mathcal{D}_{\text{test}}$ consists of $m$ data points $\{(X_i, Y_i)\}_{i=n+1,\ldots,m}$, but these $Y_i$ are the *unknown* targets that we try to predict.

**The data model:**  The training set $\mathcal{D}_{\text{train}}$ and test set $\mathcal{D}_{\text{test}}$ are generated by a two-phase process. Let $\Delta(\mathcal{X}^{\text{cat}})$ be the space all probability distributions defined over $\mathcal{X}^{\text{cat}}$, and $\mathcal{Q}$ be a probability distribution over $\Delta(\mathcal{X}^{\text{cat}})$. $\mathcal{D}_{\text{train}}$ is generated by repeating the following $k_1$ times: in the $i$-th iteration, we first take a sample $\mathcal{P}_i \in \Delta(\mathcal{X}^{\text{cat}})$ from $\mathcal{Q}$ ($\mathcal{P}$ is a distribution over $\mathcal{X}^{\text{cat}}$). Then, we sample a group of $h$ training data points, each with its categorical feature value being $c_i \in \mathcal{X}^{\text{cat}}$ and its $d$ numerical feature values being an i.i.d., sample from $\mathcal{P}$. Hence, all these $h$ training data $X_{ih+1}, \ldots, X_{(i+1)h}$ have the same categorical value $c_i \in \mathcal{X}^{\text{cat}}$, and they form the $i$-th group $G_i$ (e.g., the set of transactions of the $i$-th user). $\mathcal{D}_{\text{train}}$ contains $k_1$ groups $G_1, \ldots, G_{k_1}$. For the $i$-th data point $(X_i, Y_i)$, we use $\mathbf{g}(i)$ to denote the index of the group that contains $(X_i, Y_i)$. Hence, $(X_i, Y_i) \in G_{\mathbf{g}(i)}$. Let $\mathcal{F}$ be the hypothesis class and the true hypothesis is $f^* : \mathcal{X}^{\text{num}} \times \mathcal{X}^{\text{cat}} \to \mathcal{Y}$ such that $Y_i = f^*(x_{i1}, \ldots, x_{id}, Z_i)$ where $Z_i = \mathbb{E}_{X \sim \mathcal{P}_{\mathbf{g}(i)}}[X]$ ($Z_i$ is a $d$-dimensional vector) (e.g., the target depends not only on the numerical feature values of this particular transaction, but also the mean of the statistics of the user).

The test dataset $\mathcal{D}_{\text{test}}$ is generated in the same way and it contains $k_2$ groups $G_{k_1+1}, \ldots, G_{k_1+k_2}$. We assume the categorical feature values $c_1, \ldots, c_{k_1}, c_{k_1+1} \ldots, c_{k_1+k_2}$ are all distinct (e.g., a user id that appears in the test set does not appear in the training set).

**Feature generation:**  Here we are interested in features generation using operations such as GroupByThenMean. In our setting, the groupby operation is based on the categorical feature $\mathcal{X}^{\text{cat}}$. So, for a data point $X_i$, we can generate a set of new features $\hat{X}_i$ which can be computed from all data points in the same group $G_{\mathbf{g}(i)}$. Formally, let $\mathcal{H}$ be the set of possible feature generation function and the new feature $\hat{X}_i$ is computed as follows:

$$\hat{X}_i = H(G_{\mathbf{g}(i)}) \text{ for some } H \in \mathcal{H}.$$

For a feature generation function $H$ and predictor $f$, the loss on the data point $(X_i, Y_i)$ is

$$L(H, f, X_i, Y_i)] = \|Y_i - f(X_i, \hat{X}_i)\|^2 = \|Y_i - f(X_i, H(G_{\mathbf{g}(i)}))\|^2.$$

Our goal is to find a feature generation function $H \in \mathcal{H}$ and a predictor $\hat{f} \in \mathcal{F}$ such that the test loss is minimized

$$L_{\mathcal{D}_{\text{test}}}(H, \hat{f}) = \mathbb{E}_{(X_i, Y_i) \in \mathcal{D}_{\text{test}}} [L(H, \hat{f}, X_i, Y_i)] = \frac{1}{m} \sum_{(X_i, Y_i) \in \mathcal{D}_{\text{test}}} \|Y_i - f(X_i, H(G_{\mathbf{g}(i)}))\|^2.$$

If we do not use any feature generation, the loss of the predictor $f'$ [5] is simply

$$L_{\mathcal{D}_{\text{test}}}(f') = \mathbb{E}_{(X_i, Y_i) \in \mathcal{D}_{\text{test}}} [\|Y_i - f'(X_i)\|^2]$$

---

[5] Note that the input dimension for the predictor $f'$ here is different since there is no generated feature. But this is not an issue for our following argument.

In the following, we show that under nature conditions, we can achieve vanishing test loss by using feature generation (as the number of training samples $n$ and the minimum size of each group $h$ become larger). See Theorem 3. But if we only use the raw feature $X_i$ for predicting $Y_i$, the expected test loss of any predictor is at least a positive constant. See Theorem 4.

LEARNABILITY WITH FEATURE GENERATION

For a particular feature generation function $H$, we use $\text{Rad}_k(\mathcal{F})$ to denote the empirical Rademacher complexity of $\mathcal{F}$ over $k$ random samples:

$$\text{Rad}_k(\mathcal{F}) = \mathop{\mathbb{E}}_{\{(X_i, Y_i)\}_{i=1}^k} \frac{1}{k} \mathop{\mathbb{E}}_{\sigma} \left[ \sup_{f \in \mathcal{F}} \sum_{i=1}^k \sigma_i L(H, f, X_i, Y_i) \right]$$

where $\sigma = (\sigma_1, \cdots, \sigma_k)$ are independent Rademacher variables and $X_i$ is an i.i.d. sample from $\mathcal{P}_i$, which is an i.i.d. sample from $\mathcal{Q}$, and $Y_i = f^*(X_i, Z_i)$ where $Z_i = \mathbb{E}_{X \sim \mathcal{P}_i}[X]$. We assume that $\text{Rad}_{k_1}(\mathcal{F}) \to 0$ as $k_1 \to \infty$ (for many hypothesis classes, $\text{Rad}_{k_1}$ scales as $O(\sqrt{1/k_1})$) (Mohri et al., 2018). We also assume any function $f(\cdot, \cdot) \in \mathcal{F}$ is Lipschitz on $z$: There exists constant $C_\mathcal{F}$ such that $|f(X, Z_1) - f(X, Z_2)| \le C_\mathcal{F} \|Z_1 - Z_2\|$ for any $z_1, z_2, x \in \mathcal{X}$ and $f \in \mathcal{F}$. We further assume there exists constant $B_\mathcal{X}$ such that $\sup_{x \in \mathcal{X}} \|x\| \le B_\mathcal{X}$.

**Theorem 3.** *There is a feature generation function $H$, such that the test loss of the empirical risk minimizer $\hat{f}$ can be bounded by*

$$L_{\mathcal{D}_{test}}(H, \hat{f}) \le 2\text{Rad}_{k_1}(\mathcal{F}) + \sqrt{2\ln(4\delta^{-1})/k_1} + 6 B_\mathcal{X} C_\mathcal{F} \sqrt{2d\ln(4d(k_1 + k_2)\delta^{-1})/h}$$

*with probability at least $1 - \delta$. In particular, the test loss approaches to 0 when $k_1, h \to \infty$.*

*Proof.* We fix the feature generation function to be GroupByThenMean. In concrete, we have $\hat{X}_i = H(G_{\mathbf{g}(i)}) = \frac{1}{|G_{\mathbf{g}(i)}|} \sum_{j \in G_{\mathbf{g}(i)}} X_j$. The algorithm then find $f$ that minimizes the emprical risk:

$$\hat{f} = \arg\min_{f \in \mathcal{F}} L_{\mathcal{D}_{train}}(H, f).$$

According to Hoeffding's inequality and union bound for each dimension, with probability at least $1 - \delta/(2(k_1 + k_2))$, we have

$$\|\hat{X}_i - Z_i\| = \left\| \frac{1}{|G_{\mathbf{g}(i)}|} \sum_{j \in G_{\mathbf{g}(i)}} X_j - \mathop{\mathbb{E}}_{X \sim \mathcal{P}_{\mathbf{g}(i)}}[X] \right\| \le B_\mathcal{X} \sqrt{2d\ln(4d(k_1 + k_2)\delta^{-1})/h},$$

for some concrete $\mathbf{g}(i)$. By applying union bound on all groups, this statement holds for all $i$ with probability at least $1 - \delta/2$.

In this case, for any function $f \in \mathcal{F}$, we have

$$\left| (f(X_i, Z_i) - Y_i)^2 - (f(X_i, \hat{X}_i) - Y_i)^2 \right|$$

$$\le |f(X_i, Z_i) - f(X_i, \hat{X}_i)| \cdot \left| f(X_i, Z_i) - Y_i + f(X_i, \hat{X}_i) - Y_i \right|$$

$$\le 2 B_\mathcal{X} C_\mathcal{F} \sqrt{2d\ln(4d(k_1 + k_2)\delta^{-1})/h} \tag{2}$$

where the last inequality holds since $|f(X_i, Z_i) - f(X_i, \hat{X}_i)| \le C_\mathcal{F} \|\hat{z}_i - z_i\|$ and $\mathcal{Y} \subseteq [0, 1]$. Define $H^*$ be the function $H^*(G_{\mathbf{g}(i)}) = Z_i = \mathbb{E}_{X \sim \mathcal{P}_{\mathbf{g}(i)}}[X]$. We note this statement further implies $\left| L_{\mathcal{D}_{train}}(H^*, f) - L_{\mathcal{D}_{train}}(H, f) \right| \le \sqrt{2d\ln(4d(k_1 + k_2)\delta^{-1})/h}$ and $\left| L_{\mathcal{D}_{test}}(H^*, f) - L_{\mathcal{D}_{test}}(H, f) \right| \le \sqrt{2d\ln(4d(k_1 + k_2)\delta^{-1})/h}$ according to the definition of the loss function.

Note that each data point is not an i.i.d. sample (since two points in one group are correlated), we need to define the following group Rademacher complexity to bound the generalization error. We define *group Rademacher complexity* to be Rademacher complexity but defined over groups:

$$\text{Rad}_{k_1}^G(\mathcal{F}) := \mathop{\mathbb{E}}_{\mathcal{D}_{train}} \frac{1}{k_1} \mathop{\mathbb{E}}_{\sigma} \left[ \sup_{f \in \mathcal{F}} \sum_{i=1}^{k_1} \sigma_i \cdot \frac{1}{h} \sum_{j \in G_i} L(H, f, X_j, Y_j) \right],$$

where $\sigma = (\sigma_1, \cdots, \sigma_{k_1})$ are independent Rademacher variables. Note that if we view each group as one random sample, these groups are i.i.d. samples. Hence, we can apply the classical generalization bound via Rademacher complexity (Mohri et al., 2018), which asserts that with probability at least $1 - \delta/2$ for any function $f \in \mathcal{F}$ the following holds

$$\left| L_{\mathcal{D}_{\text{test}}}(H, f) - L_{\mathcal{D}_{\text{train}}}(H, f) \right| \leq 2\text{Rad}_{k_1}^G(\mathcal{F}) + \sqrt{2\ln(4\delta^{-1})/k_1}. \tag{3}$$

Moreover, one can see the group Rademacher complexity can be upper bounded by the ordinary Rademacher complexity for i.i.d. samples:

$$\text{Rad}_{k_1}^G(\mathcal{F}) \leq \frac{1}{h} \sum_{i=1}^{h} \mathop{\mathbb{E}}_{\mathcal{D}_{\text{train}}} \frac{1}{k_1} \mathop{\mathbb{E}}_{\sigma} \left[ \sup_{f \in \mathcal{F}} \sum_{j=1}^{k_1} \sigma_j \cdot L(H, f, X_{G_{i,j}}, Y_{G_{i,j}}) \right] = \text{Rad}_{k_1}(\mathcal{F}) \tag{4}$$

where $G_{i,j}$ is the $j$-th element in $G_i$.

According to union bound, both equation 2 and equation 3 are satisfied for all $f \in \mathcal{F}$ with probability at least $1 - \delta$. In this case, since $f^*$ is the ground truth, its empirical risks satisfies $L_{\mathcal{D}_{\text{train}}}(H^*, f^*) = 0$. In this case, we have

$$\begin{aligned} L_{\mathcal{D}_{\text{train}}}(H, \hat{f}) &\leq L_{\mathcal{D}_{\text{train}}}(H, f^*) \\ &\leq \left| L_{\mathcal{D}_{\text{train}}}(H, f^*) - L_{\mathcal{D}_{\text{train}}}(H^*, f^*) \right| + L_{\mathcal{D}_{\text{train}}}(H^*, f^*) \\ &\leq 2B_{\mathcal{X}} C_{\mathcal{F}} \sqrt{2d\ln(4d(k_1 + k_2)\delta^{-1})/h} \end{aligned} \tag{5}$$

where the first inequality dues to empirical risk minimization and the last inequality dues to equation 2. As a result, the population loss can then be bounded by

$$\begin{aligned} L_{\mathcal{D}_{\text{test}}}(H, \hat{f}) &\leq \left| L_{\mathcal{D}_{\text{test}}}(H, \hat{f}) - L_{\mathcal{D}_{\text{test}}}(H^*, \hat{f}) \right| + \left| L_{\mathcal{D}_{\text{test}}}(H^*, \hat{f}) - L_{\mathcal{D}_{\text{train}}}(H^*, \hat{f}) \right| \\ &\quad + \left| L_{\mathcal{D}_{\text{train}}}(H^*, \hat{f}) - L_{\mathcal{D}_{\text{train}}}(H, \hat{f}) \right| + L_{\mathcal{D}_{\text{train}}}(H, \hat{f}) \\ &\leq 2B_{\mathcal{X}} C_{\mathcal{F}} \sqrt{2d\ln(4d(k_1 + k_2)\delta^{-1})/h} + 2\text{Rad}_{k_1}^G(\mathcal{F}) + \sqrt{2\ln(4\delta^{-1})/k_1} \\ &\quad + 2B_{\mathcal{X}} C_{\mathcal{F}} \sqrt{2d\ln(4d(k_1 + k_2)\delta^{-1})/h} + 2B_{\mathcal{X}} C_{\mathcal{F}} \sqrt{2d\ln(4d(k_1 + k_2)\delta^{-1})/h} \\ &\leq 2\text{Rad}_{k_1}(\mathcal{F}) + \sqrt{2\ln(4\delta^{-1})/k_1} + 6B_{\mathcal{X}} C_{\mathcal{F}} \sqrt{2d\ln(4d(k_1 + k_2)\delta^{-1})/h} \end{aligned}$$

where the second inequality holds due to equation 3 and equation 5 while the last inequality follows from equation 4. This proves the desired statement. $\square$

**Remarks:** We only consider the case where $\mathcal{H}$ only contain one particular operation GroupByThenMean. In fact, it is possible to extend the above setting to one with multiple feature generation functions. Here we require that such feature generation functions be statistics of the corresponding group and can be estimated using i.i.d. samples. In our two-phase generative process, each group contains exactly $h$ data points. We can easily extend our theorem to the setting where the size of each group is also a random variable as long as it takes value at least $h$ with high probability. The main idea is the same but the notations would become very tedious. Since our goal here is to illustrate the statistical advantage of feature generation, we choose to present a simplified yet representative setting.

### WITHOUT FEATURE GENERATION

**Theorem 4.** *In case that we do not use any feature generation, there exists a problem instance such that, no matter how large $k_1$ (number of groups in the training set), $k_2$ ((number of groups in the test set)), and $h$ (the size of each group) are, for any function $f' : \mathcal{X} \to \mathcal{Y}$, the test loss is at least*

$$L_{\mathcal{D}_{test}}(f') \geq \frac{3}{64}.$$

*Proof.* Consider a problem instance with $\mathcal{X} = \mathcal{Y} = [0, 1]$. The data distribution is generated in the following way: Each group contains $h$ data points and is generated in the following way: The

distribution of $i$-th group $\mathcal{P}_{\mathtt{g}(i)}$ is random between $B(\frac{3}{4})$ and $B(\frac{1}{4})$ with equal probability where $B(p)$ is the Bernoulli distribution such that $\Pr[B(p) = 1] = p = 1 - \Pr[B(p) = 0]$. Thus, $Z_i = \mathbb{E}_{X \sim \mathcal{P}_{\mathtt{g}(i)}}[X]$ is either $\frac{3}{4}$ or $\frac{1}{4}$ with equal probability for each group. $X_i$ is 0/1 random variable, generated i.i.d. from either $B(\frac{3}{4})$ if $Z_i = \frac{3}{4}$ and from either $B(\frac{1}{4})$ if $Z_i = \frac{1}{4}$. We assume the target of data point is given by $Y_i = f^*(X_i, Z_i) = Z_i$.

When $X_i = 1$, the probability of $Y_i = \frac{1}{4}$ is given by

$$\Pr\left[Y_i = \frac{1}{4}\middle|X_i = 1\right] = \Pr\left[Z_i = \frac{1}{4}\middle|X_i = 1\right] = \frac{\Pr[Z_i = \frac{1}{4}, X_i = 1]}{\sum_z \Pr[Z_i = z]\Pr[X_i = 1|Z_i = z]} = \frac{1}{4}.$$

Similarly, the probability of $Y_i = \frac{3}{4}$ is $\Pr[Y_i = \frac{3}{4}|X_i = 1] = \frac{3}{4}$. So for any predictor $f' : \mathcal{X} \to \mathcal{Y}$, we have

$$\mathbb{E}_{X_i=1}[\|Y_i - f'(X_i)\|^2] = \frac{1}{4} \cdot \left(f'(X_i) - \frac{1}{4}\right)^2 + \frac{3}{4} \cdot \left(f'(X_i) - \frac{3}{4}\right)^2 \geq \frac{3}{64}.$$

The last inequality holds because the left hand side is a quadratic function. With the same reasoning, we have $\mathbb{E}_{X_i=0}[\|Y_i - f'(X_i)\|^2] \geq \frac{3}{64}$. As a result, the test loss of for any predictor $f'$ is at least

$$L_{\mathcal{D}_{\text{test}}}(f') = \mathbb{E}_{(X_i,Y_i)\in\mathcal{D}_{\text{test}}}[\|Y_i - f'(X_i)\|^2] \geq \frac{3}{64}.$$

The above argument does not depend on how large $k_1$, $k_2$ and $h$ are. $\qquad\square$

## B  DATA

Table 6: Datasets description

| Name | Abbr | # Train | # Validation | # Test | # Num | # Cat | # Ord | Task type |
|---|---|---|---|---|---|---|---|---|
| California Housing | CA | 13209 | 3303 | 4128 | 7 | 0 | 1 | Regression |
| Microsoft | MI | 723412 | 235259 | 241521 | 111 | 0 | 25 | Regression |
| Diabetes | DI | 65129 | 16283 | 20354 | 3 | 34 | 10 | Binclass |
| Nomao | NO | 22465 | 6000 | 6000 | 34 | 29 | 55 | Binclass |
| Vehicle | VE | 60000 | 18528 | 20000 | 100 | 0 | 0 | Binclass |
| Jannis | JA | 53588 | 13398 | 16747 | 54 | 0 | 0 | Multiclass |
| Covertype | CO | 371847 | 92962 | 116203 | 9 | 0 | 45 | Multiclass |

We describe the details of the datasets in Table 6. All the datasets can be found in the supplementary materials.

## C  ADDITIONAL RESULTS

Table 7: Comparison between OpenFE and other baselines. The results of baseline methods are from the corresponding papers. Our results are averaged by 10 different random seeds.

| Dataset | Source | C\ R | Instances\Features | Random | TransGraph | LFE | NFS | OpenFE |
|---|---|---|---|---|---|---|---|---|
| Airfoil | UCIrvine | R | 1503\5 | 0.753 | 0.801 | - | 0.796 | **0.808** |
| German Credit | UCIrvine | C | 1000\24 | 0.655 | 0.724 | - | 0.805 | **0.815** |
| Higgs Boson Subset | UCIrvine | C | 50000\28 | 0.699 | 0.729 | 0.68 | 0.731 | **0.741** |
| Ionosphere | UCIrvine | C | 351\34 | 0.934 | 0.941 | 0.932 | 0.969 | **0.986** |
| SpamBase | UCIrvine | C | 4601\57 | 0.937 | **0.961** | 0.947 | 0.955 | **0.961** |
| SpectF | UCIrvine | C | 467\44 | 0.748 | 0.788 | - | 0.876 | **0.877** |
| Sonar | UCIrvine | C | 208\60 | 0.723 | - | 0.801 | 0.839 | **0.929** |

### C.1 COMPARISONS WITH OTHER BASELINES

We compare OpenFE with other baselines, including some learning-based methods that lack critical details for code reproduction. These mehtods include:

- **Random** (Khurana et al., 2018). Randomly include features from candidate feature set multiple times and select new features with improvement according to CV scores.
- **TransGraph** (Khurana et al., 2018). TransGraph uses reinforcement learning to traverse a transformation graph for feature transformations.
- **LFE** (Nargesian et al., 2017). LFE recommends feature transformations by meta learning approaches.
- **NFS** (Chen et al., 2019). NFS uses a recurrent neural network controller to search for a series of transformations.

Following previous studies (Chen et al., 2019), the metric for regression datasets is $1 -$ (relative absolute error) and the metric for classification datasets is F1-score. We present the results in Table 7. OpenFE also surpasses other baseline methods in these datasets.

Table 8: Comparisons between the results of reproduced methods and results from the paper. The results are averaged by 10 different random seeds.

|  | Metric | FCTree (from paper) | FCTree (reproduced) |
|---|---|---|---|
| Transfusion | Accuracy | 0.752 | $0.793 \pm 0.017$ |
| Nuclear(All) | AUC | 0.629 | $0.625 \pm 0.009$ |

(a) Comparison between the results of FCTree (from paper) and FCTree (reproduced).

|  | Metric | SAFE (from paper) | SAFE (reproduced) |
|---|---|---|---|
| Magic | AUC | 0.9288 | $0.9370 \pm 0.0009$ |
| Spambase | AUC | 0.9846 | $0.9837 \pm 0.0012$ |

(b) Comparison between the results of SAFE (from paper) and SAFE (reproduced).

|  | Metric | AutoCross (from paper) | AutoCross (reproduced) |
|---|---|---|---|
| Bank | AUC | 0.9455 | $0.9456 \pm 0.0008$ |
| Adult | AUC | 0.9280 | $0.9251 \pm 0.0003$ |
| Credit | AUC | 0.8567 | $0.8624 \pm 0.0002$ |

(c) Comparison between the results of AutoCross (from paper) and AutoCross (reproduced).

### C.2 REPRODUCTION

We reproduce FCTree, SAFE, and AutoCross according to the descriptions in their papers. In order to make sure that we reproduce a reasonable version of these methods, we compare the results of our reproduced methods with the ones in their paper. We present the results in Table 8. We can see that most of the results of the reproduced methods closely match or even outperform the results from the papers.

### C.3 RUNNING TIME

We present the running time of different methods in Table 9. The experimental environment is the same for all the methods. We can see that OpenFE is a fast method that terminates within a reasonable amount of time even for large datasets.

Table 9: The running time of different methods in minutes. IEEE: IEEE-CIS Fraud Detection, BNP: BNP Paribas Cardif Claims Management.

|          | CA  | MI   | DI  | NO  | VE  | JA  | CO   | IEEE | BNP |
|----------|-----|------|-----|-----|-----|-----|------|------|-----|
| FCTree   | 2.3 | 1978 | 6.8 | 11  | 74  | 40  | 160  | -    | -   |
| SAFE     | -   | -    | 0.9 | 1.3 | 10  | -   | -    | -    | -   |
| AutoFeat | 0.2 | 23   | 37  | 49  | 535 | 354 | 1484 | -    | -   |
| AutoCross| -   | -    | 169 | 148 | 146 | -   | -    | -    | -   |
| OpenFE   | 0.1 | 31   | 4   | 4.7 | 3.3 | 3.5 | 20   | 92   | 0.9 |

Table 10: Comparisons between MDI, permutation, and SHAP in feature importance attribution.

|      | MDI | permutation | SHAP |
|------|-----|-------------|------|
| Rank | $2.07_{\pm 0.93}$ | $1.79_{\pm 0.91}$ | $2.14_{\pm 0.69}$ |
| Time | 0s | 25min | 42s |

## C.4 COMPARING FEATURE IMPORTANCE ATTRIBUTION METHODS

In this section, we compare MDI, permutation, and SHAP in feature importance attribution in OpenFE. We use the results of OpenFE on seven benchmarking datasets to rank these methods. The training model is LightGBM. We present the results in Table 10. We also present the running time of each method on the Microsoft dataset, the benchmarking dataset with largest number of samples. Different feature attribution methods do not differ much on most of the datasets. MDI can be obtained for free after the training process of LightGBM, while permutation and SHAP may require longer running time, depending on the sizes of datasets.

## C.5 KAGGLE COMPETITION

We present the details of the BNP Paribas Cardif Claims Management competition in this section. The goal of the competition is to predict whether a personal insurance claim should be approved. The competition's 8th place team made public their generated features after the competition ended[6] (the winners with higher rankings did not share their codes). We evaluate the performance of OpenFE in a similar way. A baseline model using Catboost (Prokhorenkova et al., 2018) without feature generation ranks at 31 among 2920 teams. The baseline model with features generated by experts ranks at 12/2920, while the baseline model with features generated by OpenFE ranks at 12/2920. We present the results in Table 4. OpenFE generates 200 first-order features and 100 second-order features, while Expert generates 156 first-order features and 132 high-order features.

Table 11: The results of validating feature boosting. The number is the reduction in RMSE on the validation set. FB: feature boosting.

| base feature | $bf^{(1)}$ | $bf^{(2)}$ | $bf^{(13)}$ | $bf^{(4)}$ | $bf^{(5)}$ | $bf^{(6)}$ | $bf^{(7)}$ | $bf^{(8)}$ |
|--------------|-----------|-----------|------------|-----------|-----------|-----------|-----------|-----------|
| without FB   | 0.3256 | 0.0132 | 0.1008 | 0.0041 | 0.0011 | 0.0525 | 0.1441 | 0.1852 |
| with FB      | -0.0002 | -0.0000 | -0.0002 | -0.0002 | -0.0002 | -0.0000 | -0.0001 | -0.0000 |
| synthetic feature | $sf^{(1)}$ | $sf^{(2)}$ | $sf^{(3)}$ | $sf^{(4)}$ | $sf^{(5)}$ | $sf^{(6)}$ | $sf^{(7)}$ | $sf^{(8)}$ |
| with FB      | 0.0037 | 0.0029 | 0.0053 | 0.0052 | 0.0026 | 0.0028 | 0.0026 | 0.0022 |

---

[6]https://www.kaggle.com/code/confirm/xfeat-cudf-lightgbm-catboost-wip

### C.6 VALIDATE FEATURE BOOSTING

We conduct experiments to validate that feature boosting can estimate the incremental performance of new features. We show that:

- Features that are not effective in the presence of base features have zero or negative reduction in loss $\Delta$ when evaluated using feature boosting. For example, base features are not effective in addition to themselves, and they should have zero or negative $\Delta$.
- Features that are effective in the presence of base features have positive reduction in loss $\Delta$ when evaluated using feature boosting. For example, a synthetic feature generated using the information in the targets should have positive $\Delta$.

We perform the experiment on the CA dataset ($\mathcal{D}$) with 8 base features. We generate base predictions $\hat{y}$ by the 8 base features for feature boosting. The experiments include:

- For each base feature, we train a GBDT $f'$ using the single base feature. We calculate $\Delta' = L(\emptyset) - L(f')$ of each feature. We present the results in the second row of Table 11. We show through this experiment that each base feature can indeed explain the targets.
- For each base feature $bf$, we calculate $\Delta = \text{FeatureBoosting}(\mathcal{D}, bf, \hat{y})$. We present the results in the fourth row of Table 11. We can see that all of the base features have a zero or negative $\Delta$.
- We generate 8 synthetic features using the information in the targets. The formula is $f = 0.3 \times y + \epsilon$, where $\epsilon$ follows the normal distribution with the mean and standard deviation the same as $y$. For each synthetic feature $sf$, we calculate $\Delta = \text{FeatureBoosting}(\mathcal{D}, sf, \hat{y})$. We present the results in the fifth row of Table 11. The 8 synthetic features are effective in the presence of base features, and they have positive $\Delta$ in feature boosting.

## D IMPLEMENTATION DETAILS

### D.1 EXPERIMENTAL ENVIRONMENT

For all the experiments, feature generation is carried out on a workstation with Intel(R) Xeon(R) Gold 6230 CPU @ 2.10GHz, 40 cores, 512G memory. Model tuning and model training are performed on one or more NVidia Tesla V100 16Gb.

Table 12: Unary operators.

| Numerical | Categorical |
|---|---|
| Freq, Abs, Log, Sqrt, Sigmoid, Round, Residual | Freq |

Table 13: Binary operators. (N refers to Numerical, C refers to Categorical)

| N × N | N × C | C × C |
|---|---|---|
| Min, Max, $+, -, \times, \div$ | GroupByThenMin, GroupByThenMax, GroupByThenMean, GroupByThenMedian, GroupByThenStd, GroupByThenRank | Combine, CombineThenFreq, GroupByThenNUnique |

### D.2 OPERATORS AND FEATURE TRANSFORMATIONS

All the operators are classified into unary operators (Table 12) and binary operators (Table 13), where unary operators act on one feature and binary operators act on two features. Then the operators are

further classified according to the type of features they act on. For example, GroupByThenMean requires a categorical feature and a numerical feature, while Max requires two numerical features. All the features in a dataset are classified into numerical features, categorical features, and ordinal features. The difference between ordinal features and categorical features is that an ordinal feature has a clear ordering of categories (such as "age"). Ordinal features are treated as both numerical and categorical when generating features transformations. For example, we can calculate GroupByThenMean(age, gender), which is the average age of each gender. We can also calculate GroupByThenMean(income, age), which is the average income of people of different ages. For anonymized datasets where the meanings of features are unknown, features with string values are treated as categorical features. Features with discrete values (the number of unique values is less than 100) are treated as ordinal features. Features with continuous values are treated as numerical features.

We enumerate all the first-order transformations to form the candidate feature set. A first-order transformation uses one operator once to transform base features. For example, weight/height is a first-order transformation of base features weight and height. $\text{BMI} = \text{weight/height}^2$ is a second-order transformation.

We list the details of all the operators below. $f$ stands for a numerical feature and $c$ stands for a categorical feature.

- $\text{Freq}(f)$. The frequency of feature $f$.
- $\text{Freq}(c)$. The frequency of feature $c$.
- $\text{Abs}(f)$. Element-wise absolute value.
- $\text{Log}(f)$. Element-wise logarithm.
- $\text{Sqrt}(f)$. Element-wise square root.
- $\text{Sigmoid}(f)$. Element-wisely apply function $x \mapsto \frac{1}{1+e^{-x}}$.
- $\text{Round}(f)$. Element-wise rounding.
- $\text{Residual}(f)$. Element-wisely take decimal part.
- $\text{Min}(f_1, f_2)$. Element-wise minimum.
- $\text{Max}(f_1, f_2)$. Element-wise maximum.
- $f_1 + f_2$. Element-wise addition.
- $f_1 - f_2$. Element-wise subtraction.
- $f_1 \times f_2$. Element-wise multiplication.
- $f_1 \div f_2$. Element-wise division.
- $\text{GroupByThenMin}(f, c)$. The minimum value of $f$ in each category of feature $c$.
- $\text{GroupByThenMax}(f, c)$. The maximum value of $f$ in each category of feature $c$.
- $\text{GroupByThenMean}(f, c)$. The average value of $f$ in each category of feature $c$.
- $\text{GroupByThenMedian}(f, c)$. The median of $f$ in each category of feature $c$.
- $\text{GroupByThenStd}(f, c)$. The standard deviation of $f$ in each category of feature $c$.
- $\text{GroupByThenRank}(f, c)$. The ranking of $f$ in each category of feature $c$. The rankings are normalized between 0 and 1.
- $\text{Combine}(c_1, c_2)$. Comebine the categories in feature $c_1$ and $c_2$ to be new categories. For example, for a data point, "vocation" is "doctor" and "hobby" is "football", then the value for $\text{Combine}(\text{vocation}, \text{hobby})$ of this data point is a category of "doctor-football".
- $\text{CombineThenFreq}(c_1, c_2)$. Same as $\text{freq}\,(\text{Combine}(c_1, c_2))$.
- $\text{GroupByThenNUnique}(c_1, c_2)$. The number of unique values of $c_1$ in each category of feature $c_2$.

### D.3  FEATURE BOOSTING

We describe the main idea of feature boosting in Section 3.2. Here, we present the details of the implementation in this section. The first step in feature boosting is to generate base predictions. In this paper, we use LightGBM to model tabular data and use 5-fold cross-validation to generate base predictions. For each fold, we first train a LightGBM on the train set till early stopping on the validation set. Then we generate predictions on the validation set. The predictions on each fold are concatenated to yield the base predictions. The LightGBM is trained using a set of default parameters for all the datasets. The default parameter has 10,000 estimators, 0.1 learning rate, 31 leaves, and 200 early stopping rounds. Other parameters follow the default settings in LightGBM.

The second step in feature boosting is to find a new model $f' \in F'$ to boost the performance of base predictions and minimize $L(f + f') = \sum_{i=1}^{n} l(y_i, f(\boldsymbol{x}_i) + f'(\boldsymbol{x}_i))$. In the implementation, we use the base predictions $\{f(\boldsymbol{x}_i), i = 1, 2, ..., n\}$ as the initial predictions in LightGBM. The implementation of LightGBM can use new features $\mathcal{T}'$ as the inputs to continue training from the base prediction. The new model $f'$ is the new LightGBM model after training. Feature boosting can also be extended to using neural networks. If $f'$ is a neural network parameterized by $\theta$, we can calculate $\frac{\partial L(f+f')}{\partial \theta}$ and optimize $\theta$ by back-propagation.

Table 14: The parameters of OpenFE for each dataset. IEEE: IEEE-CIS Fraud Detection. BNP: BNP Paribas Cardif Claims Management.

|                     | CA  | MI  | DI  | NO  | VE  | JA  | CO  | IEEE | BNP  |
|---------------------|-----|-----|-----|-----|-----|-----|-----|------|------|
| # data blocks       | 1   | 64  | 8   | 32  | 16  | 1   | 8   | 32   | 8    |
| # new features      | 10  | 10  | 10  | 10  | 10  | 50  | 50  | 600  | 200  |
| # candidate features | 275 | 78k | 8k  | 81k | 31k | 9k  | 27k | 456k | 1753 |

### D.4  PARAMETERS

We present the hyperparameters of OpenFE for each dataset in Table 14. For datasets whose samples or the number of candidate features are not large, the number of data blocks for successive featurewise pruning is set to be small. When the number of data blocks is one, we use all the data to calculate and evaluate new features in successive featurewise pruning, and eliminate new features with negative reduction in loss. For most datasets, the number of new features is set to be 10. The effective new features are usually sparse in the vast pool of candidate new features. For two multi-classification datasets, the number of new features is set to be 50. For a fair comparison, all the baseline methods include the same number of new features, except for the **NN** baseline. For the **NN** baseline, the number of new features is the number of hidden units in the last hidden layer, which is determined by hyperparameter tuning.

In OpenFE, the default parameter of LightGBM in successive featurewise pruning has 1000 number of estimators, 0.1 learning rate, 16 leaves, and 3 early stopping rounds. The default parameter in feature importance attribution is the same except for 50 early stopping rounds.

### D.5  FEATURE IMPORTANCE ATTRIBUTION METHODS

MDI is the gain importance embedded in LightGBM. For permutation feature importance, we randomly shuffle the values of features on the validation set, and observe the change in validation loss. For SHAP feature importance, we average the SHAP values of each sample for each feature.

## E  DISCUSSION

### E.1  DISCOVERED FEATURE TRANSFORMATIONS

In this section, we present and explain the useful transformations discovered by OpenFE for the IEEE-CIS Fraud Detection competition and the DI (Diabetes) dataset. The goal of the IEEE competition is to predict whether an online transaction is fraudulent. The top-1 feature discovered by

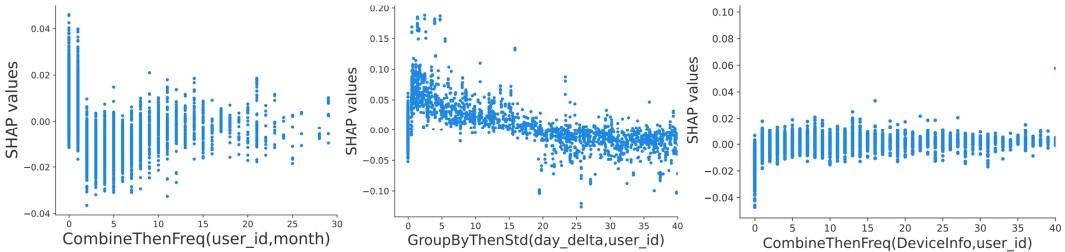

Figure 3: The SHAP values of discovered feature transformations. The x-axis is the feature values, and the y-axis is the SHAP values. Larger SHAP values indicate higher probability of fraudulent transactions.

OpenFE is $\mathrm{CombineThenFreq}(\mathrm{user\_id}, \mathrm{month})$, which indicates how many transactions a user makes in a month. We present the SHAP values of the feature in the trained XGBoost in Figure 3. The SHAP values tell us that when the user makes few transactions in a month, the transactions are more likely to be fraudulent. The top-2 feature is $\mathrm{GroupByThenStd}(\mathrm{day\_delta}, \mathrm{user\_id})$, which is the standard deviation of the days between the current transaction and the first transaction. The SHAP values imply that when many transactions happen in a short period (which corresponds to a small standard deviation), the transactions are more likely to be fraudulent. The top-4 feature is $\mathrm{CombineThenFreq}(\mathrm{DeviceInfo}, \mathrm{user\_id})$, which indicates how frequently the user switches devices to make transactions. The actual meanings are masked for most of the features to protect the privacy of users. One of the advantages of OpenFE is the ability to generate useful features without knowing their actual meanings.

The DI (diabetes) dataset collects 10 years (1999-2008) of clinical care at 130 US hospitals, where the goal is to predict the readmission (Strack et al., 2014). One can see from Table 2 that OpenFE outperforms other baseline methods and improves the learning performance by a great margin. The top-1 feature discovered by OpenFE is freq(patient_id), which is the number of times the patient has been admitted to the hospital and is highly predictive of whether the patient will be readmitted to the hospital. However, other methods may fail to find this new feature since the feature patient_id itself is not useful. For example, SAFE (Shi et al., 2020) only generates candidate features based on the base features that are used as splits in XGBoost. However, since patient_id itself is not useful, it will not be used for splits in XGBoost. This example also demonstrates that, reducing the number of candidate features by heuristic assumptions may risk missing useful candidate features.

### E.2 HIGH-ORDER FEATURES

How to search for high-order feature transformations is challenging in automated feature generation due to the explosion in search space (Chen et al., 2019). Some previous methods argue that high-order features are useful by directly searching for high-order features (Chen et al., 2019; Khurana et al., 2018). However, the effectiveness of high-order feature transformations should be evaluated in light of all its low-order components. For example, a second-order feature transformation $f_1 \times f_2 \times f_3$ is effective only if it has additional effectiveness to all their first-order components $f_1 \times f_2$, $f_1 \times f_3$, and $f_2 \times f_3$. In addition, because high-order feature transformations are typically more difficult to interpret than low-order ones, low-order feature transformations are favoured in industrial applications where interpretability is important. In a word, searching for high-order features in a hierarchical manner is more appropriate than directly searching for high-order features in two aspects: 1) evaluating the effectiveness of high-order features in a more accurate way. 2) generating low-order features with better interpretability.

Whether it is necessary to generate high-order features is case-by-case. Because the search space of high-order features is usually incredibly huge (even if we limit the order), none of the existing methods can enumerate all the high-order features within reasonable computational resources. We can hardly claim that high-order features are not useful for a dataset. However, we do not find that generating high-order features is useful for all the benchmarking datasets in our experiments. In the IEEE competition, generating high-order features does not seem to be useful for neither Expert nor OpenFE. In the BNP competition, generating high-order features provides a small improvement in

the test score. We may conclude that first-order features are usually more important than high-order ones in feature generation.

# F    COMPLEXITY ANALYSIS

**Complexity of Generating Base Predictions.** Let $n$ be the number of samples, $m$ be the number of base features of dataset, and $k$ be the number of folds. Complexity of generating base predictions is $k$ times of GBDT[7] training,

$$O\left(kTdnm\right)$$

where each GBDT contains $T$ trees with a maximum depth $d$.

**Complexity of STAGE I.** Suppose we split the dataset into $2^q$ data blocks, the number of candidate features is $m^2$. There are $2^{-q}m^2$ features remaining after successive featurewise halving. The complexity is $\sum_{i=0}^{q} 2^{(i-q)} \cdot m^2 \cdot C(2^{-i}n, 1)$, where $C(n, m)$ is the complexity of GBDT training with data shape $n \times m$. If the GBDT contains $T_1$ trees with a maximum depth of $d_1$, we have the time complexity

$$O\left(2^{-q}qT_1d_1nm^2\right)$$

**Complexity of STAGE II.** The complexity of stage II is dominated by a single GBDT training. Suppose the GBDT has $T_2$ trees with a maximum depth of $d_2$, then the time complexity of stage II is

$$O\left(2^{-q}T_2d_2nm^2\right)$$

**Overall Complexity.** In the implementation of OpenFE, the number of trees and the maximum depth of GBDT are predefined fixed constant. If we regard $T$ and $d$ as constant, the overall complexity of OpenFE is

$$O(2^{-q}qnm^2)$$

---

[7]In complexity analysis, we refer to the implementation of LightGBM. The dominant term in complexity is building histogram, i.e., $O(nm)$ per depth per tree.

