# OpenReview forum: "OpenFE: Automated Feature Generation beyond Expert-level Performance"
_ICLR.cc/2023/Conference — Submitted to ICLR 2023_

### Official Review · Reviewer_kN28 · 2022-10-21

**Confidence:** 3
**Correctness:** 4
**Technical Novelty And Significance:** 3
**Empirical Novelty And Significance:** 3
**Recommendation:** 8

**Clarity, Quality, Novelty And Reproducibility:**

- The approach itself combines and adapts already existing methods to the setting of feature engineering. To the best of my knowledge the exact use and application of these methods is novel.
- While the language of the paper is mostly clear and quite readable, Section 4 could benefit from some proofreading. "has benefit provably.", "Due to space limit", "one may think each training data as a transaction", "[...] (each user may have many transactions in this table. These transactions form a group).".
- The variables max_order in Algorithm 1 and q in Algorithm 2 are not defined.
- For permutation feature importance a wrong reference is given: (Breiman, 2001). In this paper nothing is stated about permutation feature importance.
- Bold claims are made about outperforming human experts. However, but the pub. rank is never 1 in the results which I interpret as that there exist expert approaches that perform better and human experts are not outperformed. It is not clearly stated how this conclusion is drawn.
- Several parts of the paper are outsourced to the appendix. If I counted correctly the appendix is referenced 19 times throughout the paper. For me as a reader, this makes the paper quite hard to follow and I wonder whether a 9-page conference paper is the right format for this work.


**Strength And Weaknesses:**

[+] Interesting and novel approach to automated feature engineering.
[+] In the (quite limited) empirical study the proposed approach seems to perform very well.

[-] While the proposed approach is indeed very intriguing and it seems to be a reasonable approach (also with respect to observed performance), the paper falls short on providing theoretical justifications for the made design choices. More specifically, the proposed method is only described as it is but not much is stated about why the two stage approach was followed nor was there some motivation on the concrete techniques employed.

[-] The empirical study is quite limited. In total, 7 datasets were considered plus some Kaggle competitions. While such a small number of datasets might be enough to demonstrate the existence of advantages, it is not enough to state whether it really outperforms all the state-of-the-art methods.

[-] Moreover, since OpenFE is a heuristic, it most likely will have some shortcomings. No such shortcomings are observed nor discussed. In particular, no limitations of the approach are discussed throughout the paper.

[-] Another open question is on base of which datasets the method was developed. If it was with respect to the evaluated datasets there might be a positive bias in the data.

[-] Furthermore, it was not stated how the datasets were picked and why other datasets were not considered.


**Summary Of The Paper:**

The paper proposes an approach to automated feature generation which is split into two subroutines. After new candidate features are generated, in the first subroutine the set of newly generated features is pruned via an adaptation of successive halving which is dubbed successive pruning. In the second stage a boosting approach is followed based on the effect of a feature to reduce the loss of a learner.


**Summary Of The Review:**

To sum up, I have the overall impression that the paper is not yet ready for publication, especially not at such a prestigious venue as ICLR. While the proposed method is reasonable and the overall idea is novel, the paper falls short on the justifications for design decisions, scope of the experimental study, and discussion of limitations. Significant parts are outsourced to the appendix making the overall content hard to grasp. All in all, I recommend to reject the paper but I would like to encourage the authors to further pursue their work and revise the paper since the paper would make a great contribution also with the open source implementations of other approaches.


Edit after rebuttal: My concerns vanished after the authors gave their rebuttal along with more experimental results. However, I am still missing some discussion of the limitations which might be hard to spot due to the outstanding performance.

---

> ### Author Response · Authors · 2022-11-10
> **Response to Reviewer kN28**
>
> Thank you for the detailed review!
>
> > For permutation feature importance a wrong reference is given: (Breiman, 2001). In this paper nothing is stated about permutation feature importance.
>
> Please refer to the last paragraph of page 23 in (Breiman, 2001):
>
> > After each tree is constructed, the values of the mth variable in the out-of-bag examples are randomly permuted and the out-of-bag data is run down the corresponding tree.’
>
> exactly describes permutation feature importance. Now, (Breiman, 2001) has been regarded as the source of permutation feature importance and many subsequent works (e.g.  [1, 2, 3]) cite (Breiman, 2001) when mentioning permutation feature importance.
>
> > The empirical study is quite limited.
>
> > There might be a positive bias in the data.
>
> > Why other datasets were not considered.
>
> We address these concerns by providing more experiments on 20 datasets. Please refer to the global response. These datasets are unbiased because they were gathered and used by other papers. We hope that OpenFE's superior performance on these datasets will address your concerns about the empirical results. In our own opinion, most of these datasets used by previous studies [4, 5, 6, 7, 8] only contain a few hundreds to a few thousand data points with a few dozens of features, which is the reason we did not perform evaluation on them in the previous verison. Experimental results on toy datasets are hardly convincing for the real progress in feature generation. The datasets used in our paper range in size from moderate to big, with some having millions of samples and hundreds of features. What's more, our results on Kaggle competitions strongly support OpenFE's effectiveness, not only because these Kaggle competitions are highly competitive, but also because we show that the features generated by OpenFE outperform the features generated by the winning team (out of 6351 teams).
>
> > Section 4 could benefit from some proofreading.
>
> > The variables max_order in Algorithm 1 and q in Algorithm 2 are not defined.
>
> We have polished and modified the paper according to your advice on the language and undefined variables.
>
> > Bold claims are made about outperforming human experts.
>
> Please refer to the global response https://openreview.net/forum?id=CnG8rd1hHeT&noteId=CIZq7DAUYvP. We conducted FAIR comparisons to demonstrate that OpenFE's features outperform those of a winning team in a highly competitive Kaggle competition, which definitely supports our claim that ‘OpenFE CAN outperform human experts in feature generation.’
>
> ***
> ### Reference
>
> [1] Hooker G, Mentch L, Zhou S. Unrestricted permutation forces extrapolation: variable importance requires at least one more model, or there is no free variable importance[J]. Statistics and Computing, 2021, 31(6): 1-16.
>
> [2] Zhou Z, Hooker G. Unbiased measurement of feature importance in tree-based methods[J]. ACM Transactions on Knowledge Discovery from Data (TKDD), 2021, 15(2): 1-21.
>
> [3] Covert I, Lundberg S M, Lee S I. Understanding global feature contributions with additive importance measures[J]. Advances in Neural Information Processing Systems, 2020, 33: 17212-17223.
>
> [4] Katz G, Shin E C R, Song D. Explorekit: Automatic feature generation and selection[C]//2016 IEEE 16th International Conference on Data Mining (ICDM). IEEE, 2016: 979-984.
>
> [5] Chen X, Lin Q, Luo C, et al. Neural feature search: A neural architecture for automated feature engineering[C]//2019 IEEE International Conference on Data Mining (ICDM). IEEE, 2019: 71-80.
>
> [6] Zhu G, Xu Z, Yuan C, et al. DIFER: differentiable automated feature engineering[C]//International Conference on Automated Machine Learning. PMLR, 2022: 17/1-17.
>
> [7] Khurana U, Samulowitz H, Turaga D. Feature engineering for predictive modeling using reinforcement learning[C]//Proceedings of the AAAI Conference on Artificial Intelligence. 2018, 32(1).
>
> [8] Nargesian F, Samulowitz H, Khurana U, et al. Learning Feature Engineering for Classification[C]//Ijcai. 2017: 2529-2535.

---

### Official Review · Reviewer_mToT · 2022-10-22

**Confidence:** 3
**Correctness:** 3
**Technical Novelty And Significance:** 3
**Empirical Novelty And Significance:** 1
**Recommendation:** 6

**Clarity, Quality, Novelty And Reproducibility:**

The overall writing of this paper is clear but a little redundant, since the method is mentioned too many times before being officially introduced. Still, the framework has been well explained and supported by the experiments. Moreover, the mathematical derivation is complex but logical.

**Strength And Weaknesses:**

The main strengths of this paper are as follows:
1. The author has done sufficient experiments, to illustrate the effectiveness of the proposed method. Also, by reproducing the main methods of this task, this paper provides convenience for future work.
2. This paper gives theoretical proof for the advantage of feature generation over the base feature set.
3. The proposed method is well explained and the overall logic is smooth.

The main weaknesses of this paper are as follows:
1. The method proposed in this paper is still restricted to the existing expand-and-reduce framework, so the innovation of the overall method is limited. One possible optimization direction is to eliminate part of the features when constructing the candidate set.
2. The writing of this article is a little redundant. As a result, the important ablation study section has to be put in the appendix.
3. In the experiments section, the definition of the experts is not clear enough. Especially when claiming to outperform human experts, the exact definition of the comparison object needs to be specified.


**Summary Of The Paper:**

This paper proposes an evaluation method to eliminate redundant candidate features following the expand-and-reduce framework for automated feature generation. The proposed method achieves state-of-the-art performance on seven benchmarks and outperforms human exports for the first time. Moreover, a theoretical analysis is provided to prove the advantage of feature generation over the base feature, strengthening this task's importance and facilitating future research.

**Summary Of The Review:**

The method proposed in this paper is not the most innovative, but the authors contribute to this field by theoretically proving the advantage of feature generation. Overall, the paper is marginally below the acceptance threshold of ICLR2023.

======================================================================================================
Thanks for the responses. My main concerns have been answered and resolved by the authors, including the completeness of datasets, the overall clarity, the complexity and runtime of OpenFE, and a detailed explanation of the Kaggle competition. Also, according to the supplementary results on 20 datasets, OpenFE achieves a better overall performance than FETCH. Considering the authors’ response, the contributions of open-source implementations, and some issues that did exist in the first submission, I have decided to change my recommendation to 6: marginally above the acceptance threshold.

---

> ### Author Response · Authors · 2022-11-10
> **Response to reviewer mToT**
>
> Thank you for the review!
>
> > The method proposed in this paper is still restricted to the existing expand-and-reduce framework, so the innovation of the overall method is limited.
>
> We do not agree with the reasoning that adopting an expand-and-reduce framework results in little innovation. In fact, practically all of the current feature generation methods adhere to the expand-and-reduce framework, i.e., they need to first expand the candidate features, and then remove redundant ones [1, 2, 3, 4, 5, 6]. Our innovation lies in that: 1) we propose a novel feature boosting method to efficiently and accurately evaluate the incremental performance of new features, 2) we propose an evaluation framework with successive featurewise halving and feature importance attribution to quickly eliminate redundant features. The sub-components of OpenFE can be readily incorporated into existing feature generation methods.
>
> > One possible optimization direction is to eliminate part of the features when constructing the candidate set.
>
> Thank you for the nice suggestions. There are some previous works that tried this optimization direction. For instance, SAFE [1] first trains an XGBoost model and then only constructs candidate feature sets using features on the same path in the XGBoost model. The basic assumption of SAFE is that uninformative features (that are not used as splits in XGBoost) do not yield informative candidate features after transformation. This assumption is not true in many datasets and may result in the deletion of important candidate features. In a Diabetes dataset, for example, when the goal is to forecast if a patient will be readmitted to the hospital, the feature 'patient id' is useless. However, 'freq(patient id),' which is the number of times the patient has been admitted to the hospital, is a strong predictor of whether the patient would be readmitted.
>
> > The writing of this article is a little redundant. As a result, the important ablation study section has to be put in the appendix.
>
> Thanks for your advice. We made an effort to polish our paper and try to minimize redundancy. The main body of the paper now includes the ablation study.
>
> > In the experiments section, the definition of the experts is not clear enough.
>
> Although it is difficult to define experts precisely, we compare the features generated by OpenFE with the feature generated by the winning (1st ranked) team (out of 6351 teams) in the IEEE-CIS Fraud Detection Competition. Kaggle competitions are highly competitive. Participants in the top 1% should undoubtedly be regarded machine learning experts (in order to get in top 1%, it is necessary to use sophisticated feature engineering and various model training tricks), not to mention the winning team.
>
> ***
> ### Reference
> [1] Shi Q, Zhang Y L, Li L, et al. Safe: Scalable automatic feature engineering framework for industrial tasks[C]//2020 IEEE 36th International Conference on Data Engineering (ICDE). IEEE, 2020: 1645-1656.
>
> [2] Katz G, Shin E C R, Song D. Explorekit: Automatic feature generation and selection[C]//2016 IEEE 16th International Conference on Data Mining (ICDM). IEEE, 2016: 979-984.
>
> [3] Chen X, Lin Q, Luo C, et al. Neural feature search: A neural architecture for automated feature engineering[C]//2019 IEEE International Conference on Data Mining (ICDM). IEEE, 2019: 71-80.
>
> [4] Zhu G, Xu Z, Yuan C, et al. DIFER: differentiable automated feature engineering[C]//International Conference on Automated Machine Learning. PMLR, 2022: 17/1-17.
>
> [5] Khurana U, Samulowitz H, Turaga D. Feature engineering for predictive modeling using reinforcement learning[C]//Proceedings of the AAAI Conference on Artificial Intelligence. 2018, 32(1).
>
> [6] Fan W, Zhong E, Peng J, et al. Generalized and heuristic-free feature construction for improved accuracy[C]//Proceedings of the 2010 SIAM International Conference on Data Mining. Society for Industrial and Applied Mathematics, 2010: 629-640.

---

### Official Review · Reviewer_YEEM · 2022-11-04

**Confidence:** 2
**Correctness:** 3
**Technical Novelty And Significance:** 2
**Empirical Novelty And Significance:** 3
**Recommendation:** 5

**Clarity, Quality, Novelty And Reproducibility:**

# Clarity
Although the paper is well written, the proposed OpenFE method is not described in sufficient detail within the main paper. Similarly, the experimental results are difficult to put into context, as the baselines and the extent to which they resemble OpenFE are not sufficiently described (are there baselines that also generate new features? Which baselines simply consider pruning the feature space)?

# Quality
Experimental results across a variety of datasets showcase that OpenFE outperforms the considered baselines by a significant amount, while remaining comparatively computationally cheap (cf. Table 8).

# Novelty
This works combines previous work into separate steps that make up OpenFE. Although this is not as of itself a bad thing, the novelty could potentially be improved by further theoretical analysis and worst-case bounds as discussed earlier.

# Reproducibility
The authors provide the source code for OpenFE.

**Strength And Weaknesses:**

# Strengths
- The paper is well written, and addresses a crucial problem in real-world ML applications.
- Experimental results across several datasets show that OpenFE outperforms competing feature generation baselines, such as DCN-v2 and TabNet.
- The authors consider a variety of ablation experiments to verify the importance of the different OpenFE components.
- The authors reimplement and open-source baselines considered in this paper.

# Weaknesses
- I believe that the main weakness of this work lies in its overall clarity. The authors provide a high-level description of OpenFE, but spend too little time discussing the specifics. For example,
   + How important is it to increase the highest order of potential features in OpenFE? How does this trade off with the overall runtime?
   + Similarly, the authors provide a theoretical analysis of when feature generation can provably improve model performance. However, the assumptions that are made about the training data - crucial to the analysis and overall scope of the result - are left entirely to the appendix. I would recommend providing a quick summary of the assumptions and why they are required for the proof in the main paper.
- The authors do not provide an analysis of the overall runtime of OpenFE, despite the potential size of the explored feature space and resulting algorithms being major impediments to automated feature extractions. I recommend the authors include some formal analysis, including the complexity of OpenFE, as well as (ideally) some guarantees on the two feature refinement subprocesses. Would it be possible to state anything in terms of how well Algorithm 2 approximates its optimal output?
- The authors compare their works on a Kaggle competition, but focus on comparing to methods that did not rank first in that competition. Although this may be reasonable (algorithmic constraints, etc.), doing so must be justified. What is the difference between the model that placed 42nd and the model that placed first? Do I understand correctly that they were proposed by the same team?

# Questions for the authors
- Could you confirm that in Table 3, the OpenFE results describe Gradient-Boosted Trees with augmented features learned with OpenFE (but without using expert features)?
- Could sub-components of OpenFE also be combined with existing feature generation methods?

**Summary Of The Paper:**

The authors introduce a feature generation and model boosting methodology, dubbed "OpenFE".

OpenFE proceeds by
1. Generating a set of possible features, by ways of atomic operations (elementary operations, as well as "grouped" operations that apply mean/min/max operations over subsets of the data corresponding to the same categorical feature).
2. Pruning the bulk of redundant features based on (Jamieson & Talwalkar, 2016).
3. Ranking remaining features by their relative importance (the top k acting as the final set of extracted features).

The authors verify OpenFE on seven datasets, as well as by comparing OpenFE to competitors on a past Kaggle benchmark.


**Summary Of The Review:**

This is an interesting paper, but the overall clarity of the proposed method is insufficient in the paper's current state. Although empirical results show the value in OpenFE, it is difficult to put them in context.

---

> ### Author Response · Authors · 2022-11-10
> **Response to Reviewer YEEM**
>
> Thank you for the detailed review!
>
> > How important is it to increase the highest order of potential features in OpenFE?
>
> We discussed high-order features in Appendix E.2. The main conclusion is that, for the majority of datasets, producing first-order features suffices, and we have not seen evidence that higher-order features can further enhance learning performance. This conclusion is also supported by the outcomes of Kaggle competitions, where the performance can rarely be improved by high-order features created by human specialists. If you are interested in more discussions, please refer to Appendix E.2.
>
> > The authors do not provide an analysis of the overall runtime of OpenFE, despite the potential size of the explored feature space and resulting algorithms being major impediments to automated feature extractions.
>
> The overall running time of OpenFE is provided in Table 8. We provide the size of the explored search space in Table 14. According to your advice, we include a formal analysis on the complexity of OpenFE in Appendix F.
>
> > The authors compare their works on a Kaggle competition, but focus on comparing to methods that did not rank first in that competition.
>
> Please refer to the global review https://openreview.net/forum?id=CnG8rd1hHeT&noteId=CIZq7DAUYvP. We DID make a direct comparison between OpenFE's features and that of the first ranked team.
>
> > Could you confirm that in Table 3, the OpenFE results describe Gradient-Boosted Trees with augmented features learned with OpenFE (but without using expert features)?
>
> Yes, the OpenFE results in Table 3 and 4 only have augmented features discovered by OpenFE.
>
> > Could sub-components of OpenFE also be combined with existing feature generation methods?
>
> Yes. In general, most feature generations methods need to: 1) evaluate the incremental performance of a new feature, 2) eliminate redundant candidate features. In OpenFE, we propose a novel feature boosting method to efficiently and accurately evaluate the incremental performance of new features. Moreover, we propose an evaluation framework to quickly eliminate redundant candidate features through successive featurewise halving and feature importance attribution. These components can be readily incorporated into existing feature generation methods.

---

### Author Response · Authors · 2022-11-10
**Global response to the reviewers (1/3)**

We would like to thanks all reviewers for insightful and helpful comments.

***

## More empirical evaluations
We notice that another concurrent submission to ICLR2023 on the topic of automated feature generation titled ‘Learning a Data-Driven Policy Network for Pre-Training Automated Feature Engineering’ (FETCH) (https://openreview.net/forum?id=688hNNMigVX) receives scores of 8, 8, 8. We have evaluated our OpenFE on the benchmark datasets used in their paper and find out that our OpenFE beats FETCH on most datasets as well. Their code does not compile successfully (missing important source files) and the scores for FETCH and other methods in this table are excerpted from their paper.

| | #samples | #features | Base | DFS | AutoFeat | NFS | DIFER | FETCH | OpenFE |
| :-: | :-: | :-: | :-: | :-: | :-: | :-: | :-: | :-: | :- |
| Airfoil | 1503 | 5 | 0.5068 | 0.6003 | 0.5955 | 0.6226 | 0.6125 | 0.6463 | **0.7894$_{\pm1.2e-2}$** |
| BikeShare DC | 10886 | 11 | 0.9880 | 0.9990 | 0.9891 | 0.9991 | 0.9995 | 0.9997 | **0.9998$_{\pm1.8e-5}$** |
| Housing Boston | 506 | 13 | 0.4641 | 0.4708 | 0.4703 | 0.4977 | 0.5072 | 0.5224 | **0.6867$_{\pm2.2e-2}$** |
| House King County | 21613 | 19 | 0.6843 | 0.6908 | 0.6917 | 0.6934 | 0.6948 | **0.7475** | 0.7359$_{\pm8.1e-3}$ |
| Openml_586 | 1000 | 25 | 0.6564 | 0.7188 | 0.7178 | 0.7223 | 0.6946 | 0.7671 | **0.7746$_{\pm1.4e-2}$** |
| Openml_589 | 1000 | 25 | 0.6395 | 0.6959 | 0.7278 | 0.7165 | 0.6789 | **0.7562** | 0.7488$_{\pm1.5e-2}$ |
| Openml_607 | 1000 | 50 | 0.6363 | 0.6815 | 0.6499 | 0.6485 | 0.6564 | 0.7404 | **0.7518$_{\pm1.6e-2}$** |
| Openml_616 | 500 | 50 | 0.5605 | 0.5807 | 0.5927 | 0.5856 | 0.5982 | 0.6749 | **0.6894$_{\pm2.4e-2}$** |
| Openml_618 | 1000 | 50 | 0.6351 | 0.6848 | 0.6374 | 0.6461 | 0.6553 | 0.7351 | **0.7472$_{\pm2.4e-2}$** |
| Openml_620 | 1000 | 25 | 0.6309 | 0.6528 | 0.6574 | 0.6943 | 0.7262 | 0.7506 | **0.7536$_{\pm2.1e-2}$** |
| Openml_637 | 500 | 50 | 0.5160 | 0.5105 | 0.5763 | 0.5739 | 0.6006 | 0.6453 | **0.6772$_{\pm1.5e-2}$** |
| Amazon Employee | 32769 | 9 | 0.9492 | 0.9447 | 0.9499 | 0.9510 | 0.9504 | 0.9516 | **0.9737$_{\pm1.2e-3}$** |
| Credit_a | 690 | 6 | 0.8044 | 0.8056 | 0.8086 | 0.8101 | 0.8108 | 0.8114 | **0.8829$_{\pm2.8e-2}$** |
| Fertility | 100 | 9 | 0.8700 | 0.7900 | 0.8910 | 0.9189 | 0.8800 | 0.8900 | **0.9288$_{\pm2.8e-2}$** |
| Hepatitis | 155 | 12 | 0.8258 | 0.8516 | 0.8677 | 0.8766 | 0.8839 | **0.9290** | 0.8862$_{\pm4.9e-2}$ |
| Messidor Features | 1150 | 19 | 0.6594 | 0.7089 | 0.7359 | 0.7417 | 0.7541 | 0.7689 | **0.7693$_{\pm3.0e-2}$** |
| SpamBase | 4601 | 57 | 0.9154 | 0.9198 | 0.9237 | 0.9341 | 0.9372 | 0.9405 | **0.9434$_{\pm7.6e-3}$** |
| SpecfF | 267 | 44 | 0.7751 | 0.8125 | 0.8331 | 0.8608 | 0.8538 | 0.8838 | **0.8911$_{\pm1.4e-2}$** |
| Wine Quality Red | 999 | 12 | 0.5597 | 0.5422 | 0.5641 | 0.5814 | 0.5779 | 0.6042 | **0.7023$_{\pm9.4e-3}$** |
| Wine Quality White | 4900 | 12 | 0.4976 | 0.4855 | 0.5023 | 0.5111 | 0.5153 | 0.5235 | **0.6866$_{\pm1.4e-2}$** |

We have carefully investigated the details of FETCH’s supplementary materials to ensure fair comparisons on the same experimental settings. In our supplementary materials, we have given the datasets and codes needed to replicate these experiments (see rebuttal.txt in the zip file). Our algorithm runs extremely fast on those datasets (less than half an hour for all of them. So one can actually verify our experimental results). We once again show the excellent ability of OpenFE in improving the learning performance of tabular data, which is the most crucial component in automated feature generation.

_(continued)_

---

> ### Author Response · Authors · 2022-11-10
> **Global response to the reviewers (2/3)**
>
> Feature engineering is a very crucial component in the learning pipeline for tabular data in many industrial applications. But the problem has received relatively less attention in research papers. As a result:
> - There is a lack of commonly used benchmark datasets for this problem. Existing approaches only evaluate their methods on toy datasets [1, 2, 3, 4, 5] (such as those listed in the table), the majority of which contain only a few hundreds to a few thousand data points with a few dozens of features. Experimental results on toy datasets are hardly convincing for the research progress in feature generation.
> - Due to a lack of efficient and general feature generation methods, existing open-source AutoML pipeline packages (such as AutoGluon [6], AutoSklearn [7], AutoWEKA [8], TPOT [9], H2O [10], AutoX [11]) either use naïve feature generation methods or simply skip feature generation in the pipeline.
>
> In order to address the problems of feature generation in both academia and industry, our paper:
> - Designs an efficient automated feature generation method that really handles large scale datasets (with million data points and hundreds features).
> - Directly tackles two well-participated Kaggle competitions, which are clearly more challenging than most of the toy datasets used in previous work [1, 2, 3, 4, 5]. More importantly, we demonstrate for the first time that OpenFE can provide competitive results against human experts, with the hope that efficient and effective feature generation methods (such as OpenFE) can be incorporated into existing AutoML pipelines, freeing machine learning experts from the laborious task of manual feature generation.
>
> Once again, we would like to highlight some features of OpenFE:
> 1. The search space of candidate features is very carefully designed so that the algorithm can efficiently discover useful features from it. It results from our prior knowledge gained through numerous empirical studies on a variety of real-world datasets (but unfortunately there is not much story we can tell in the paper about this part).
> 2. The algorithm is simple but definitely NOT straightforward. One can easily find that a full test of the effectiveness of a new feature would almost certainly necessitate training a new model with the new feature and existing features, which would be prohibitively expensive. We use several tricks to make the algorithm as efficient as possible while also keeping it simple and effective.
> 3. Our algorithm is light-weighted. It directly runs on the training dataset (without the need of any other “similar” datasets for transferring knowledge). One can even use it in a laptop (without GPU).
> 4.  We notice that FETCH receives scores of 8, 8, 8.  As one can imagine, we are quite disappointed with the current ratings of 5, 5, 3, especially given that OpenFE can significantly beat FETCH on most datasets. We firmly believe in the practical value of OpenFE and hope that machine learning experts will use it to enhance the performance of their models.
> We response to other comments from each reviewer separately below. We admit that there are some presentation issues and thrive to address them in the revised version. But we feel strongly that these  issues alone do not justify a downright rejection. Therefore, we would like to defend our work and hope that our clarification and additional experimental results can help the reviewers to better evaluate the contribution of our work.
>
> _(continued)_

---

> > ### Author Response · Authors · 2022-11-10
> > **Global response to the reviewers (3/3)**
> >
> > ## Kaggle competitions
> >
> > Some reviewers argued that we did not compare with experts that rank at 1st and human experts are not outperformed. There is a **misunderstanding** here that needs to be clarified. For the IEEE-CIS Fraud Detection, we compare OpenFE with the features generated by the competition’s first place team (see https://www.kaggle.com/code/cdeotte/xgb-fraud-with-magic-0-9600). The competition is highly competitive with 6351 data science teams participated. The first-place team only shared one of the XGBoost models they used. The actual model they used to achieve the first rank is an ensemble of many learning models (including LGBM, CAT, XGB). But the team withheld the details of these models (one can also see the comments in https://www.kaggle.com/code/cdeotte/xgb-fraud-with-magic-0-9600).
> > > When this model is ensembled together with Konstantin's CatBoost and LGBM models, the result achieves public LB 0.9677 and private LB 0.9459 taking first place.
> >
> >  However, rather than designing improved learning models, our attention is on generating effective features. We have conducted FAIR comparisons (i.e., using the same learning model) to show that the features generated by OpenFE lead to greater performance improvement than the features generated by the first-place team. Nevertheless, we agree that claiming OpenFE can outperform human experts may not be rigorous enough. We plan to change this claim to ``The features generated by OpenFE provide competitive (comparable) results against the features generated by human experts’’.
> >
> >
> > ***
> > ### References
> > [1] Katz G, Shin E C R, Song D. Explorekit: Automatic feature generation and selection[C]//2016 IEEE 16th International Conference on Data Mining (ICDM). IEEE, 2016: 979-984.
> >
> > [2] Chen X, Lin Q, Luo C, et al. Neural feature search: A neural architecture for automated feature engineering[C]//2019 IEEE International Conference on Data Mining (ICDM). IEEE, 2019: 71-80.
> >
> > [3] Zhu G, Xu Z, Yuan C, et al. DIFER: differentiable automated feature engineering[C]//International Conference on Automated Machine Learning. PMLR, 2022: 17/1-17.
> >
> > [4] Khurana U, Samulowitz H, Turaga D. Feature engineering for predictive modeling using reinforcement learning[C]//Proceedings of the AAAI Conference on Artificial Intelligence. 2018, 32(1).
> >
> > [5] Nargesian F, Samulowitz H, Khurana U, et al. Learning Feature Engineering for Classification[C]//Ijcai. 2017: 2529-2535.
> >
> > [6] Erickson N, Mueller J, Shirkov A, et al. Autogluon-tabular: Robust and accurate automl for structured data[J]. arXiv preprint arXiv:2003.06505, 2020.
> >
> > [7] Feurer M, Klein A, Eggensperger K, et al. Efficient and robust automated machine learning[J]. Advances in neural information processing systems, 2015, 28.
> >
> > [8] Thornton C, Hutter F, Hoos H H, et al. Auto-WEKA: Combined selection and hyperparameter optimization of classification algorithms[C]//Proceedings of the 19th ACM SIGKDD international conference on Knowledge discovery and data mining. 2013: 847-855.
> >
> > [9] Olson R S, Moore J H. TPOT: A tree-based pipeline optimization tool for automating machine learning[C]//Workshop on automatic machine learning. PMLR, 2016: 66-74.
> >
> > [10] LeDell E, Poirier S. H2o automl: Scalable automatic machine learning[C]//Proceedings of the AutoML Workshop at ICML. 2020, 2020.
> >
> > [11] https://github.com/4paradigm/AutoX

---

### Author Response · Authors · 2022-11-19
**Looking forward to more feedbacks**

Dear reviewers,

Thank you again for your insightful comments and advice, which are really helpful in refining the paper. We have uploaded a modification and offered responses to the concerns raised. More empirical research has also been done by us to show that OpenFE is not just incredibly efficient and outperforms FETCH in generalization performance (see https://openreview.net/forum?id=688hNNMigVX). It would be greatly appreciated if you could also look at the interesting discussions between FETCH and ourselves.

We totally understand that this is a particularly busy time, as reviewers may be preparing rebuttals for their own submissions or hurrying to meet conference deadlines.

We would greatly appreciate it if you could provide more comments on whether our responses resolve your concerns. If there are any more concerns, we'll do our utmost to address them.

Best,

The Authors

---

### Decision · Program_Chairs · 2023-01-20

**Decision:**

Reject

**Justification For Why Not Higher Score:**

The paper presents a pipeline of heuristics and existing methods for feature extraction.  While the presented empirical results seem strong, the paper lacks clarity and as such it's unclear what the novelty is, what is attributable to the existing methods used in the pipeline and reviewers seemed to not understand the method in its entirety.  As such, it doesn't seem ready for publication.

**Justification For Why Not Lower Score:**

N/A

**Metareview: Summary, Strengths And Weaknesses:**

This paper presents OpenFE, which is an automated algorithm for feature generation for tabular data.  The algorithm proceeds via two stages.  First the algorithm does "successive featurewise pruning" using a bandit approach (successive halving) followed by sensible heuristics to estimate the effectiveness of individual features and prune less effective ones.  Then there is a feature importance attribution stage that uses a variety of heuristics along with the MDI algorithm.  The authors show strong results across a variety of experiments including comparing OpenFE to experts in a Kaggle competition and a variety of related algorithms.

Initial reviews were just below borderline, with reviewers finding that method likely performs quite well in practice, but found it a bit heuristic without understanding justification for some choices.  They raised questions regarding the empirical evaluation, and particularly related to the choice of comparisons used (e.g. they didn't understand why certain Kaggle experts were compared to and others, e.g. the top ranked, were not).  A lot of questions related to clarity.  The paper describes a pipeline of algorithmic choices and it seems reviewers had trouble following what choices were made exactly and why at different stages.  In personal discussion, one reviewer remarked that from the text they wouldn't feel like they would be able to reimplement the pipeline based on the writing.  Another question relates to novelty, it's not entirely clear what the authors' contribution is and what is attributable to gradient boosted decision trees and MDI (perhaps this relates to the clarity point above).  It seems like these are the two main components of the two steps in the pipeline?

In the response phase, the authors presented additional results and in particular compared empirically to another paper in review at this conference with a higher average score.  One reviewer was compelled to raise their score to 8, based on this comparison ("It would be a pitty if the better approach would not be accepted. I increased my score to suggest acceptance accordingly.").  While such strong results should be applauded, such a comparison seems problematic.  Raw empirical performance is only one dimension of many that should be considered for acceptance.  In addition, we cannot base scores on those of another paper for which the review process is incomplete (e.g. there are comments on that paper questioning the empirical evaluation there).  Therefore, the reasoning for raising the score will be downweighted in the decision.  The reviewers other comments "the paper falls short on the justifications for design decisions, ..., and discussion of limitations. ... making the overall content hard to grasp... etc." seem to be shared by the other reviewers and should be addressed.

Therefore, the recommendation is to reject the paper.  It seems that improving the clarity of the work, and addressing other reviewer comments would go a long way to convincing reviewers for a subsequent submission.

**Summary Of Ac-Reviewer Meeting:**

I wasn't able to track down all reviewers in time, but I met virtually and discussed the paper / review with one of them.